# Hydraulic suppression of basal glacier melt in sill fjords

Johan Nilsson[1,2], Eef van Dongen[1,2], Martin Jakobsson[3,2], Matt O'Regan[3,2], and Christian Stranne[3,2]

[1]Department of Meteorology, Stockholm University, 10691 Stockholm, Sweden
[2]Bolin Center for Climate Research, Stockholm University, 10691 Stockholm, Sweden
[3]Department of Geological Sciences, Stockholm University, 10691 Stockholm, Sweden

**Correspondence:** J. Nilsson (nilsson@misu.su.se)

**Abstract.** Using a conceptual model, we examine how hydraulically-controlled exchange flows in silled fjords affect the relationship between the basal glacier melt and the features of warm intermediate Atlantic Water (AW) outside the fjords. We show that an exchange flow can be forced to transit into the hydraulic regime if the AW interface height decreases, the AW temperature increases, or the production of glacially modified water is boosted by subglacial discharge. In the hydraulic regime, the heat transport across the sill becomes a rate limiting factor for the basal melt, which is suppressed. An interplay between processes near the ice–ocean boundary and the hydraulically-controlled exchange flow determines the melt dynamics, and the sensitivity of the basal melt to changes of the AW temperature is reduced. The model results are discussed in relation to observations from Petermann, Ryder, and 79°N glaciers in North Greenland.

## 1 Introduction

In response to global warming, the Greenland Ice Sheet (GIS) has lost mass over the past decades and its marine outlet glaciers are retreating (Mouginot et al., 2019; Straneo and Heimbach, 2013; Wood et al., 2021). GIS holds an ice volume equivalent to 7.4 m of sea level (Morlighem et al., 2017), and it may contribute up to 0.3 m sea-level rise by the end of this century (Aschwanden et al., 2019). Over half of the recent mass loss from GIS is by increased discharge of ice into the ocean from marine outlet glaciers (Mouginot et al., 2019), where calving and oceanic melt of marine ice have increased. Loss of marine-terminating glaciers can cause a positive feedback: resistive stresses in grounded or floating marine glaciers buttress ice inland, and ice-stream flow speed and ice export across the grounding line increase when marine glaciers retreat (Schoof, 2007; Gudmundsson, 2013; Nick et al., 2013; Schoof et al., 2017). This accelerated ice loss contributes directly to sea-level increase.

The subsurface melt on marine glaciers is primarily controlled by the excess ocean temperature over the (pressure dependent) freezing temperature at the grounding line (Holland and Jenkins, 1999), the point where the ice begins to float (or for tidewater glaciers, the water depth at their essentially vertical fronts). The melt depends also on factors such as basal slope, subglacial discharge, tidal currents, and water column stratification (Jenkins, 2011; Truffer and Motyka, 2016; De Andrés et al., 2020). The ice melt mixes with ocean water, which creates a buoyant melt water plume that rises along the base of the ice tongue (Lewis and Perkin, 1986). Turbulence in the melt water plume transports heat to the ice–ocean boundary and sustains melt in the rising plume. Marine glaciers in Greenland terminate in fjords, and basal melt is chiefly driven by heat supplied in

subsurface Atlantic Water (AW) that enters the fjords (Straneo et al., 2012). In Greenland, basal melt is sensitive to the AW temperature (Straneo and Heimbach, 2013), and increases in AW temperature and subglacial discharge have been the major drivers of the retreat of outlet glaciers in deep Greenlandic fjords since the mid 1990s (Wood et al., 2021; Slater and Straneo, 2022). However, local features, such as fjord geometry and wind conditions, affect the sensitivity of the basal melt to changes
of the AW temperature in the open ocean (Straneo and Cenedese, 2015; Khazendar et al., 2019; Wood et al., 2021).

    The present study is motivated by recent observations of hydraulically-controlled exchange flows at sills in the Greenlandic fjords that host the ice tongues of Ryder and 79°N glaciers (Jakobsson et al., 2020; Schaffer et al., 2020). The hydraulic control sets an upper limit on the exchange flow that depends on sill geometry and upstream stratification (Pratt and Whitehead, 2007). Accordingly, hydraulic control limits the heat transport that sustains the basal melt, and has the potential to stabilise marine
glaciers. Numerous observations of sill flows demonstrate that the vertical mixing increases strongly when the flow becomes hydraulically controlled (Pratt and Whitehead, 2007), and Jakobsson et al. (2020) and Schaffer et al. (2020) show that as inflowing AW passes over the sills and descends on the landward slopes, it mixes with overlaying cold glacially-modified water. As a result, the waters reaching these glaciers' grounding lines are colder than the AW outside the fjords. This reduces the basal melt compared to the case when unmodified AW reaches the glacier. Thus, hydraulic control can reduce basal melt
by limiting the exchange flow as well as by decreasing the water temperature at the grounding line of the glacier.

    The feature that hydraulic control (and/or fjord geometry) can limit the heat transport suggests that there are two different regimes of ice-tongue basal melt in fjords. First, one where the basal melt is controlled locally by turbulent processes near the ice–ocean boundary, which determines the heat flux from the fjord water to the ice (Fig. 1a). In this scenario, the fjord-scale circulation adjusts to deliver the heat required for the basal melt, and AW reaches the grounding line of the ice tongue. Second,
in sill fjords hydraulic control may be established, which constrains the exchange circulation in the fjord and its associated heat transport (Fig. 1b). In this case, the basal melt is not solely controlled by local processes near the ice–ocean boundary, as the fjord-scale heat transport towards the glacier enters as a rate limiting factor. In the relatively narrow Greenlandic fjords, sill geometry is likely to be a major factor constraining ocean heat transport towards marine glaciers (Zhao et al., 2021; Bao and Moffat, 2023). For large Antarctic ice shelves, effects due to Earth's rotation are important and the oceanic heat flux available
for basal melt may be controlled by mesoscale ocean eddies or large-scale flows constrained by conservation of potential vorticity (Little et al., 2008; Hattermann et al., 2014; Zhao et al., 2019).

    The observations from Ryder and 79°N glaciers (Jakobsson et al., 2020; Schaffer et al., 2020) raise the question of how strongly hydraulic control limits basal melt, and how it affects the dependence of basal melt to the temperature and height of the AW layer outside the fjords. Here, we examine this question using a conceptual two-layer fjord model that includes
ocean–glacier interactions. The model results are discussed in relation to observations from the Greenlandic ice tongues of Petermann, Ryder and 79°N glaciers. However, with some modifications the model can be applied also to fjords with tidewater glaciers. Before the model is presented, we give a brief overview of the oceanographic conditions at Petermann and Ryder glaciers.

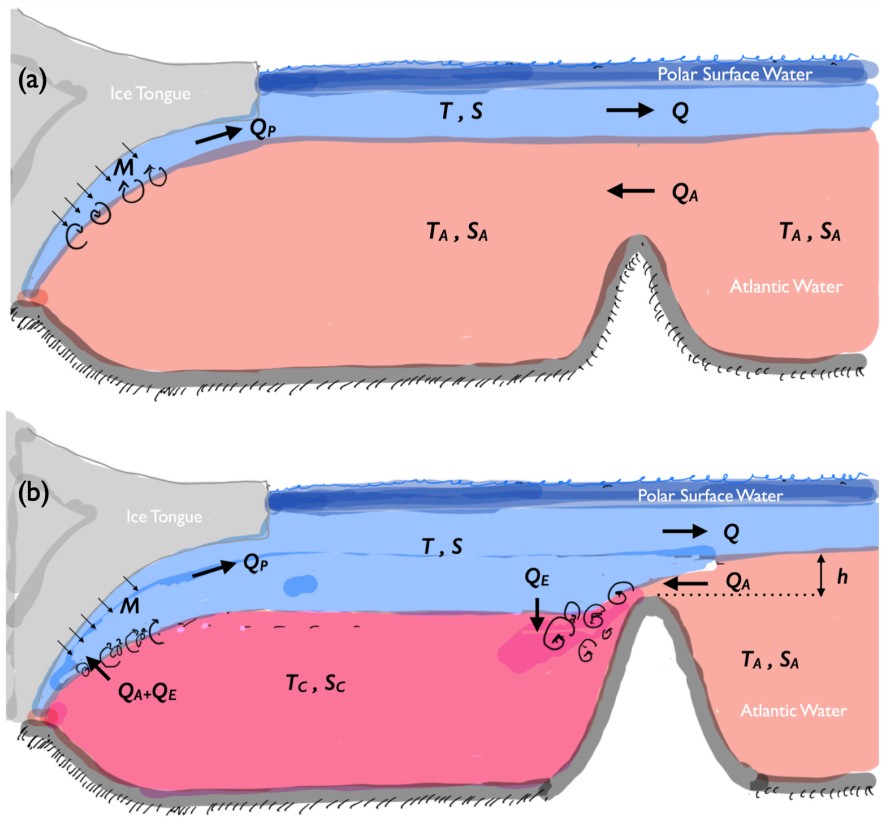

**Figure 1.** Oceanographic characteristics in Greenlandic fjords with ice tongues (or tidewater glaciers), showing the two-layer model described in section 2. Warm Atlantic Water (AW, with temperature and salinity $T_A$ and $S_A$) is encountered at depth outside the fjord. Above the AW, there is a layer of colder outflowing glacially-modified water ($T$ and $S$), which is capped by low salinity Polar Surface Water. Two flow regimes are shown: (a) a melt-controlled regime, where the exchange flow is unconstrained and AW reaches the grounding line; (b) a hydraulically-controlled regime, where outflowing water mixes with inflowing AW, thereby reducing the temperature and salinity reaching the grounding line ($T_C$, $S_C$). Model variables, listed in table 1, include AW inflow ($Q_A$), the outflow of glacial water ($Q$), entrainment into the inflowing AW ($Q_E$), plume flow at ice base ($Q_P$), and basal melt ($M$). The AW height above the sill ($h$) and the layer density difference determine the exchange flow in the hydraulically-controlled regime.

## 1.1 Ryder and Petermann glaciers

The model result will be discussed in relation to the ice tongues of 79°N, Petermann, and Ryder glaciers. These glaciers have the three largest ice tongues in Greenland (Wilson et al., 2017; Hill et al., 2018), and are located in the northern part of the island (Fig. 2). The geometries of the fjords in which these glaciers drains have some general features in common, including relatively large sill depths: about 500 m for 79°N; and about 400 m for Petermann and Ryder glaciers. Here, we will describe fjord geometries and oceanographic conditions for Petermann and Ryder, which are located relatively close (∼200 km apart)

and drain in fjords that terminate in Lincoln Sea. The oceanographic conditions in the fjord of 79°N Ice Tongue, which is Greenland's largest and about 80 km long, are described by for example Lindeman et al. (2020) and Schaffer et al. (2020).

Figures 2 and 3 shows bathymetric and temperature conditions in Sherard Osborn and Petermann fjords, where Ryder and Petermann glaciers drain. In Petermann Fjord, which has a ∼400 m deep and ∼12 km wide sill, AW with similar features are encountered inside as well as outside the fjord, and there are no indications of hydraulic control at the sill (Johnson et al., 2011;

Jakobsson et al., 2020).

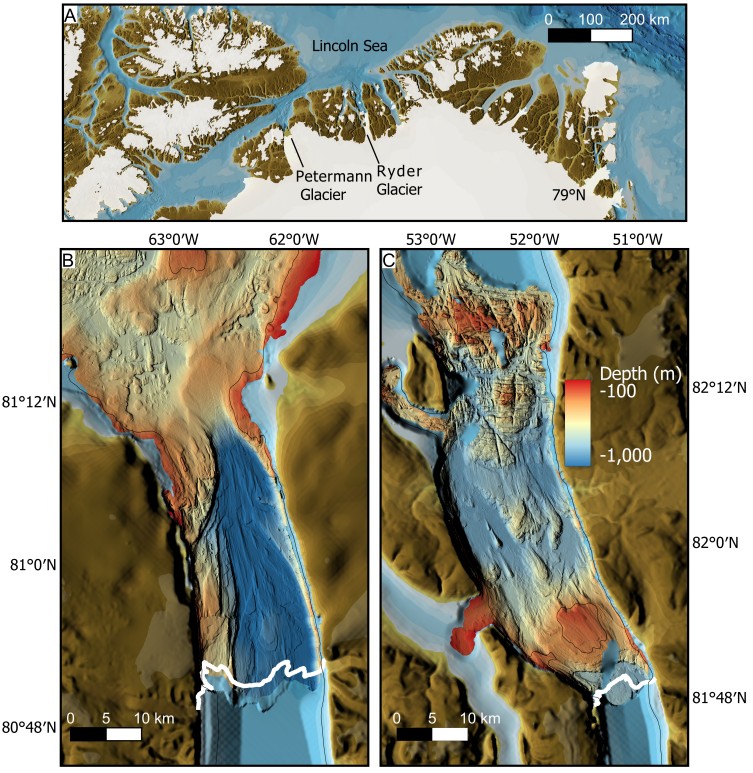

**Figure 2.** (a) Location of Petermann, Ryder and 79°N Glaciers in North Greenland. Bathymetry in Petermann Fjord (b) and in Sherard Osborn Fjord where Ryder Glaciers drain (c); the white lines indicate the frontal positions of the ice tongues in 2019. Petermann's ice tongue is about 70 km long, and about 600 (150) m thick at the grounding line (front). Ryder's ice tongue is about 30 km long, and about 700 (150) m thick at the grounding line (front).

Sherard Osborn Fjord has a more constrictive fjord topography, with an outer and an inner sill. The temperature in the AW depth range decreases across the sills, with the coldest temperature in the fjord basin landward of the inner sill that is largely capped by the ice tongue. The largest temperature drop occurs over the inner sill. Here, a strong near-bottom inflow was observed, occurring in a ∼400 m deep and ∼1 km wide channel on the eastern sill, demonstrating that the inflow is hydraulically controlled (see Fig. 4 in Jakobsson et al., 2020). Accordingly, the inner sill provides the main geometrical constraint on the exchange flow and heat transport to the glacier.

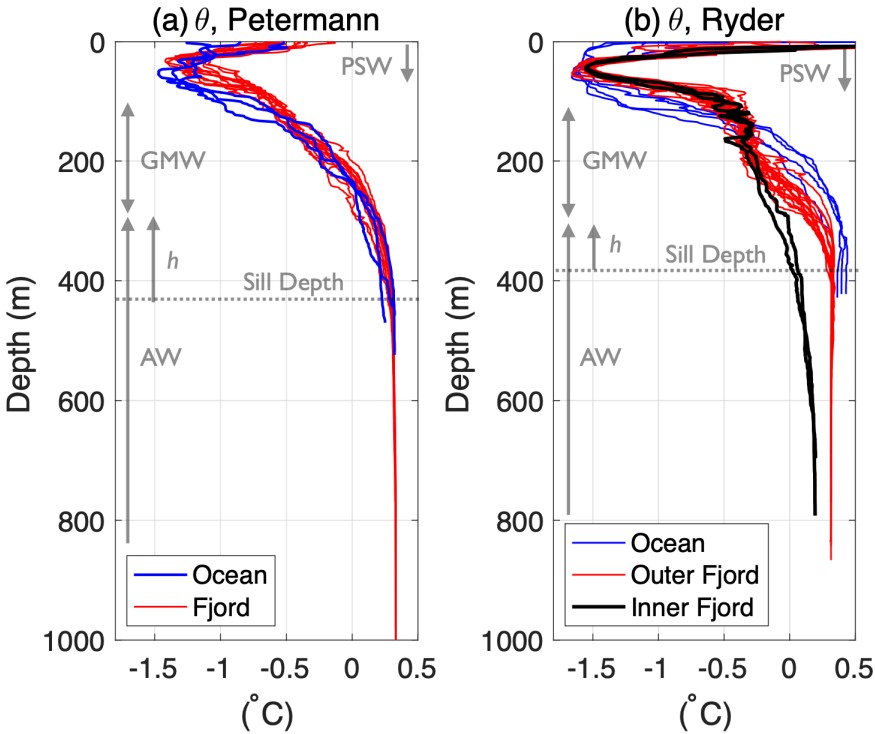

**Figure 3.** Potential temperature profiles from Petermann Fjord (a) and Sherard Osborne Fjord (b), where Ryder Glaciers drains. The observations were taken in August 2019 during the Ryder Expedition with the icebreaker *Oden*, see Jakobsson et al. (2020) and Stranne et al. (2021) for further information. Sherard Osborne Fjord has two sills: progressing into the fjord, the AW temperature drops as the sills are crossed; the red temperature profiles were taken between the two sills, and the black ones landward of the inner sill close to the ice-tongue front (Fig. 2). The horizontal dashed grey lines show the maximum sill depths; about 440 and 390 m for Petermann and Ryder, respectively. The grey vertical lines indicate approximately the vertical extents of inflowing Atlantic Water (AW, lower layer of the model) and outflowing glacially modified water (GMW, upper layer of the model), as well as Polar Surface Water (PSW) on the seaward side of the sills. The PSW layer is relatively fresh and buoyant, and prevents the GMW from reaching the sea surface. The approximate height of the AW above the sill ($h$) is also indicated.

During the last six decades the ice tongues of Petermann and Ryder have evolved differently: the former retreating significantly ($\sim$300 m per year) and the latter advancing modestly ($\sim$40 m per year) (Hill et al., 2018); and in 2010 and 2012 Petermann lost $\sim$35 km of its tongue in major calving events (Johannessen et al., 2013). The two glaciers are relatively closely located and are expected to experience similar atmospheric conditions and to have AW of similar temperatures outside the fjords. Jakobsson et al. (2020) proposed that the differences in fjord bathymetry are the reason for different behaviour of the two glaciers: Ryder Glacier has a more restrictive sill geometry, partly protecting the ice tongue from inflow of warmer subsurface AW (Fig. 2).

In summary, these results suggest tentatively that the basal melt on Petermann is chiefly rate limited by processes near the
ice–ocean boundary, whereas the basal melt on Ryder is partly rate limited by large-scale heat transport towards the glacier. In
addition to differences in sill geometry, differences in summer sea-ice conditions influence the efficiency in transporting AW
into the two fjords. Perennial land-fast sea ice outside Sherard Osborne Fjord curtail wind-driven water exchange between the
fjord and the open ocean; whereas ice-free conditions in and around Petermann Fjord allows for a more vigorous wind-driven
water exchange during summer (Shroyer et al., 2017; Jackson et al., 2018; Stranne et al., 2021). We will now go on to describe
a two-layer model that will be used to examine the interplay between basal melt dynamics and hydraulic control.

## 2  A two-layer model

We consider a two layer model of glacier–ocean interaction in a fjord, with AW ($T_A$,$S_A$) and glacially modified water ($T$,$S$)
(see Straneo and Cenedese, 2015; Jackson and Straneo, 2016, for a background). Figure 1 shows the model geometry for two
different circulation regimes that will be examined. The model represents near steady-state conditions, and we assume that
the time-mean exchange flow in the fjord is primarily driven by basal melting of the ice tongue, which creates a buoyant melt
water plume raising along its base. Higher up, the plume becomes neutrally buoyant and feeds the outflow of glacially modified
water. A fresh, low density layer of polar surface water caps the two water mass represented in the model, insulating them from
surface runoff, and contact with sea ice and the atmosphere. The polar surface water is not explicitly represented in the model.

Although subglacial discharge can have a strong impact on subsurface melt rates, we will for simplicity neglect subglacial
discharge in the model's freshwater budget. The reason is twofold. First, the resulting model becomes simpler and more
tractable analytically. In appendix A, we describe a more complex model version that includes subglacial discharge in the
conservation relations: this shows that the results remain qualitatively similar even when the subglacial discharge is significantly
greater than the subsurface melt. Second, observations indicate that freshwater input due to basal melt exceeds subglacial
discharge for large ice tongues such as 79°N, Ryder, and Petermann: Schaffer et al. (2020) estimated that in the annual mean the
subglacial discharge constitutes only about 10% of the freshwater exported from 79°N Glacier; and the summer measurements
from Petermann of Washam et al. (2019) indicate that that the freshwater fraction due to subglacial discharge in the glacially-
modified water column below the ice tongue is less than 30% (see their figure 5). This suggests that for large ice tongues,
subglacial discharge may, as a leading order approximation, be neglected in the model's freshwater budget; but subglacial
discharge will be allowed to affect the model's melt rates.

### 2.1  Conservation relations

In the two-layer model of the fjord, melting of the ice tongue is the only local source/sink of freshwater/heat. This can be used
to formulate conservation relations for volume, salt and heat (Jackson and Straneo, 2016; Truffer and Motyka, 2016). At the
sill, conservation of volume is given by

$$Q = Q_A + M, \tag{1}$$

**Table 1.** Definition of model variables and physical constants, see Fig. 1.

| | |
|---|---|
| $T_A, S_A, h$ | Atlantic Water (AW) $T$, $S$, and height |
| $T, S$ | Glacially-modified water $T$ and $S$ |
| $T_C, S_C$ | Ice cavity water $T$ and $S$ |
| $T_f = T_f(S_A, d)$ | Freezing point at grounding-line depth ($d$) |
| $\Delta T = T_A - T$ | Layer difference in $T$ |
| $\Delta S = S_A - S$ | Layer difference in $S$ |
| $\mathcal{T} = T_C - T_f$ | Thermal forcing |
| $\mathcal{T}_A = T_A - T_f$ | AW thermal forcing |
| $Q, M$ | Exchange flow and melt water input |
| $Q_A, Q_P, Q_H$ | AW, melt-plume, and hydraulic flows |
| $c \approx 4 \cdot 10^6$, $c_i \approx 2 \cdot 10^6$ | Heat capacity of water and ice (J m$^{-3}$ °C$^{-1}$) |
| $L \approx 3 \cdot 10^8$ | Latent heat of fusion (J m$^{-3}$ ) |
| $T_G = L/c + c_i/c(T_f - T_i)$ | Gade temperature ($\approx 80$ °C ) |

where $Q$ and $Q_A$ is the volume outflow and inflow of glacially modified and Atlantic waters respectively, and $M$ the freshwater input due to melting. The ice consists of pure freshwater, implying that the meltwater input $M$ does not affect the salinity budget. Hence conservation of salt is given as

$$SQ = S_A Q_A. \tag{2}$$

Combining Eqs. (1, 2) yields Knudsen's relation for salt conservation

$$\Delta S Q = S_A M, \qquad \Delta S \stackrel{\text{def}}{=} S_A - S. \tag{3}$$

The heat budget involves a balance between advective heat transport towards the glacier and basal melt. The advective heat flux is

$$H = c(T_A Q_A + T_f M - TQ), \tag{4}$$

where $c$ is the heat capacity (per unit volume) of sea water, and the melt freshwater input $M$ is assumed to have the salinity dependent freezing temperature $T_f$. Note that $T_f$ will be taken as constant set by the grounding-line pressure and $S_A$. Using Eqs. (1,4), we obtain

$$H = c[\Delta T Q + (T_f - T_A)M] \qquad \Delta T \stackrel{\text{def}}{=} T_A - T. \tag{5}$$

The heat flux is related to the ice melt ($M$) as

$$H = M[L + c_i(T_f - T_i)], \tag{6}$$

where $L$ is the latent heat of freezing, $c_i$ the heat capacity of ice, and $T_i$ the ice temperature.

By combining Eqs. (5,6), we obtain

$$\Delta T Q = M[L/c + c_i/c(T_f - T_i) + (T_A - T_f)], \tag{7}$$

Here, $L/c \approx 75\ °\text{C}$, and in North Greenlandic fjords $T_A - T_f$ is typically 3 °C. Hence, the terms involving $T_A - T_f$ can to a good approximation be neglected in Eqs. (5,7). It is convenient to define a generalised Gade temperature (Gade, 1979)

$$T_G \stackrel{\text{def}}{=} L/c + c_i/c(T_f - T_i), \tag{8}$$

which would be the decrease in temperature of a unit volume of water from which sensible heat is extracted to melt ice corresponding to a unit volume of liquid water. [Note that the equivalent ice temperature defined by Jenkins (1999) is approximately $-T_G \cdot \frac{\rho_w}{\rho_i}$, where $\frac{\rho_w}{\rho_i} \approx 1.1$ is density ratio between water and ice.] By using the definition of $T_G$, Eq. (7) can be written as

$$\Delta T Q = T_G M, \tag{9}$$

which gives the relation between the advective heat flux and the melt. Note that unless the ice temperature is very cold, $T_G \approx L/c$.

The conservation relations of the model can be summarised as follows:

1. Volume: The melt water input $M$ is small, implying that $Q \approx Q_A$. Thus, the inflow of AW approximately equals the volume outflow of glacially modified water. In what follows, we will denote the exchange flow simply by $Q$.

2. Salt: The salt balance is given by Eq. (3). Here, $M$ cannot be neglected since it is multiplied by $S_A$, which is larger than $\Delta S$: Equation (3) states that $M/Q = \Delta S/S_A$.

3. Heat: The heat budget is specified by Eq. (9), which in combination with Eq. (3) yields

$$\frac{\Delta S}{S_A} = \frac{\Delta T}{T_G}. \tag{10}$$

For melting of ice in sea water, heat and salt conservation yields a linear relationship between the salinity and temperature

differences (Gade, 1979). This allows us to either use $\Delta T$ or $\Delta S$ in our analyses; we will use $\Delta T$.

The layer density difference is calculated using a linear equation of state

$$\Delta \rho = \rho_0(\beta \Delta S - \alpha \Delta T), \tag{11}$$

where $\rho_0$ is a constant seawater reference density, and where $\alpha = 4 \cdot 10^{-5}\ \text{K}^{-1}$ and $\beta = 8 \cdot 10^{-4}$ are the thermal and haline expansion coefficients, respectively. Equation (10) allows the density difference to be written

$$\frac{\Delta \rho}{\rho_0} = \frac{\Delta T}{T_G}(\beta S_A - \alpha T_G). \tag{12}$$

Here, $(\alpha T_G)/(\beta S_A) \approx 0.1$ showing that the salinity dominates the density difference.

## 2.2 Basal melt parameterisation

We will use a parametrisation of the basal melt ($M$), which depends on the difference between the ocean water temperature ($T_C$) and the freezing point temperature of sea water ($T_f$) at the grounding line (Holland et al., 2008; Jenkins, 2011; Xu et al., 2013; Favier et al., 2019). We denote the thermal forcing as

$$\mathcal{T} \stackrel{\text{def}}{=} T_C - T_f. \tag{13}$$

If AW reaches the grounding line, then $T_C = T_A$, and the thermal forcing is denoted

$$\mathcal{T}_A \stackrel{\text{def}}{=} T_A - T_f. \tag{14}$$

However, mixing between in- and out-flowing waters over a sill can may lower $T_C$ relative to $T_A$, which implies that $\mathcal{T}$ can be lower than $\mathcal{T}_A$.

We model the area-integrated basal melt as (Xu et al., 2013)

$$M = \gamma_1 \mathcal{T}^{n_1}, \tag{15}$$

where $\gamma_1$ is a coefficient that depends on features such as the ice-tongue geometry and subglacial discharge, and $n_1 > 0$ an exponent. We assume that also the plume volume flow $Q_P$ is a function of the thermal forcing $\mathcal{T}$ and given by

$$Q_P = \gamma_2 \mathcal{T}^{n_2}, \tag{16}$$

where $\gamma_2$ is a constant and $n_2 > 0$ an exponent.

Several studies have used models of varying complexity to examine the relationship between thermal forcing and the area-integrated melt on Greenlandic ice tongues and Antarctic ice shelves (e.g. Jenkins, 1991, 2011; Little et al., 2009; Lazeroms et al., 2018, 2019; Holland et al., 2008; Cai et al., 2017; Favier et al., 2019). These investigations report values of $n_1$ in the range from 1 to 2, with a preference for $n_1$ values of around 1.5 (Xu et al., 2013; Cai et al., 2017) to 2.0 (Holland et al., 2008; Little et al., 2009). The reported range of $n_1$ is likely to reflect both different ice–ocean interaction regimes and model assumptions on the boundary conditions at the ice–ocean boundary. There are fewer studies that specifically comment on the relationship between the volume transport in the plume and the thermal forcing, but Holland et al. (2008) reported a linear dependence of $Q_P$ on the thermal forcing, i.e. $n_2 \approx 1$.

Primarily, the heat flux to the ice and the associated melting depend on the product of the thermal forcing and the plume velocity (say $u$), which in turn is related to the plume buoyancy (Holland and Jenkins, 1999; Favier et al., 2019). If the plume buoyancy is proportional to $\mathcal{T}$, and the buoyancy force is balanced by a linear basal friction, then $u \propto \mathcal{T}$. This gives $n_1 = 2$ corresponding to a quadratic relation between melt and thermal forcing (Holland et al., 2008; Little et al., 2009). If the basal friction is quadratic, i.e. proportional to $u^2$, on the other hand, scaling analyses suggest that $u \propto \mathcal{T}^{1/2}$ (Lazeroms et al., 2018), which gives $n_1 = 1.5$. Jakobsson et al. (2020) applied the plume model of Jenkins (1991) to Ryder Ice Tongue, and their results suggest that $n_1 \approx 1.7$ and $n_2 \approx 0.7$. Notably, if $M \propto u\mathcal{T}$ and $Q_P \propto u$, then $M/Q_P \propto \mathcal{T}$. In view of Eqs. (15,16) this implies

that

$$n_1 - n_2 = 1. \tag{17}$$

This constraint on the exponents leads to some attractive simplifications of the dynamics, which will be used in the analyses.

It is worth noting that theoretical considerations (e.g. Straneo and Cenedese, 2015; Jenkins, 2011, and referencers therein) indicate that on marine glaciers where the buoyancy source is dominated by the subglacial discharge near the grounding line, rather than by the distributed melt along the submerged glacier, the melt is approximately proportional to the thermal forcing, and the plume volume transport is essentially independent of the thermal forcing. This limit of high subglacial discharge, characteristic of summer conditions at Greenlandic tidewater glaciers (e.g. Straneo and Cenedese, 2015), is described by the case $n_1 = 1$ and $n_2 = 0$; which satisfies the constraint of Eq. (17). Some aspects of the case with high subglacial discharge is discussed in appendix A.

In summary, the literature reports a range of values for $n_1$ and $n_2$. However, $n_1 = 2$ and $n_2 = 1$ appear as one reasonable choice for the exponents for qualitatively examining the dynamics of large Greenlandic ice tongues such as the 79°N, Petermann and Ryder glaciers. This will be the base-line case when we examine the interplay between melt dynamics and hydraulic control in section 3. In section 3.2.5, we will consider how variations of the values of $n_1$ and $n_2$ affects the results, including the case $n_1 = 1$ and $n_2 = 0$. When we derive general results below, however, we will allow $n_1$ and $n_2$ to be arbitrary positive numbers, but subject to the constraint $n_1 > n_2$.

## 2.3 The melt-controlled exchange flow regime

Consider a situation in which the fjord geometry, via frictional resistance or hydraulic control, does not limit the exchange flow and its associated heat transport towards the ice tongue. We assume that unmodified AW reaches the glacier ($T_C = T_A$), and the melt processes create a plume volume flow $Q_P$ that sets the exchange flow: If $T_A$ increases also the exchange flow increases at the rate given by Eq. (16). This regime, in which the strength of the exchange flow is controlled locally by the glacier basal melt, will be referred to as the melt-controlled regime; and a contrasting hydraulically-controlled exchange flow regime will be presented in section 2.4.

In the melt-controlled regime, where $Q = Q_P$, we can use the heat-conservation relation (9) together with Eqs. (15,16) to obtain

$$\frac{\Delta T}{T_G} = \frac{\gamma_1}{\gamma_2} \mathcal{T}^{n_1 - n_2}. \tag{18}$$

This relationship, which is equal to $M/Q$, shows that $\Delta T$ as well as the ratio $M/Q$ increase with $\mathcal{T}$, i.e. the melt water fraction in the plume increases with thermal forcing. Note that the condition $n_1 - n_2 = 1$ yields a linear relation between $\Delta T$ as well as $M/Q$ and the thermal forcing.

By dividing Eq. (18) with $\mathcal{T}/T_G$, we obtain

$$\frac{\Delta T}{\mathcal{T}} = \frac{T_G \gamma_1}{\gamma_2} \mathcal{T}^{n_1 - n_2 - 1}. \tag{19}$$

Since $\mathcal{T} = \mathcal{T}_A$ in the melt-controlled regime, the left-hand side in this expression equals $(T_A - T)/(T_A - T_f)$; which is less or equal to one since $T \geq T_f$. When $n_1 - n_2 = 1$, the right-hand side becomes independent of $\mathcal{T}$ and equals

$$\sigma \overset{\text{def}}{=} \frac{T_G \gamma_1}{\gamma_2}. \tag{20}$$

This parameter is a non-dimensional measure of the temperature of the outflowing glacially-modified water ($T$): when $\sigma = 1$, $T = T_f$; and when $\sigma = 0$, $T = T_A$. (This interpretation of the Eq. (19) applies also when $n_1 - n_2 \neq 1$, but then $\Delta T / \mathcal{T}$ is no longer constant in the melt controlled regime.) As will be shown below, $\sigma$ influences aspects of the hydraulically-controlled regime.

To summarise, the flow in the melt-controlled regime is specified by a knowledge of the AW properties $T_A$ and $S_A$, which determine the thermal forcing $\mathcal{T} = \mathcal{T}_A$. In turn, this yields $M$, $Q$, and $\Delta T$ (Eqs. 15,16,18), and $\Delta S$ is obtained from Eq. (10).

We will now go on to examine how a hydraulically-controlled exchange flow affects the melt dynamics. For this purpose, it is useful to write $Q_P$ as a function of the temperature difference. By using Eq. (18), we obtain

$$Q_P = \gamma_2 \left( \frac{\gamma_2 \Delta T}{\gamma_1 T_G} \right)^{\frac{n_2}{n_1 - n_2}}. \tag{21}$$

Since $n_1 > 0$ and $n_2 > 0$, this shows that the melt-controlled exchange flow increases with $\Delta T$ when $n_1 - n_2 > 1$.

## 2.4 Hydraulic control

Fjord and sill geometries may impose limits on the exchange flow, which in turn can potentially alter the basal melt dynamics. In particular, hydraulic control of a two-layer exchange flow over a sill sets an upper bound for the exchange flow (say $Q_H$), which is determined by the upstream height of the AW layer above the sill ($h$) and the layer density difference (Pratt and Whitehead, 2007; Zhao et al., 2021). Exchange flow strengths below the critical value $Q_H$ are unconstrained by the geometry, and are referred to as subcritical flows. Thus, it is conceivable that a sufficiently strong melt-driven exchange flow, or a high sill, can cause a transition from a subcritical flow to a critical, hydraulically-controlled flow (Pratt and Whitehead, 2007).

How the flow evolves as the melt-driven exchange flow (or the sill height) is gradually increased and approaches the hydraulically-controlled limit is complex and depends on fjord and sill geometry (Armi, 1986; Pratt and Whitehead, 2007; Nycander et al., 2008). Observations from the Ryder and 79°N glaciers show that the inflow at the sills in front of the ice tongues are hydraulically controlled, and that the thickness of the inflow layer is thin compared to the upper outflowing layer (Jakobsson et al., 2020; Schaffer et al., 2020). This implies that the flow can be approximated by a one-layer hydraulic model representing the inflowing AW layer. Two additional features allow for simplifications of the hydraulic model. First, the depth of AW layer on the seaward side (upstream) of the sill is much larger than at the sill, which implies that the upstream AW inflow velocity is negligible. Second, the inflow over the sill is confined in a channel that is small compared to the internal Rossby radius, which allows the Earth's rotation to be neglected. In this situation, the maximum hydraulically-controlled volume flow is given by (Pratt and Whitehead, 2007)

$$Q_H = W h^{3/2} \left( \frac{2}{3} \right)^{3/2} \left( \frac{g \Delta \rho}{\rho_0} \right)^{1/2}, \tag{22}$$

where $W$ is the cross-sectional width of the lower layer on the sill[1], $h$ the height of the lower layer above the sill upstream (Fig. 1), and $g$ the gravitational acceleration. By using Eq. (12), $Q_H$ can be written as

$$Q_H = k_H h^{3/2} (\Delta T / T_G)^{1/2},  \qquad (23)$$

where we have introduced

$$k_H \overset{\text{def}}{=} W \left( \frac{2}{3} \right)^{3/2} [g (\beta S_A - \alpha T_G)]^{1/2}. \qquad (24)$$

In the hydraulically-controlled regime, the exchange flow is given by Eq. (23), i.e. $Q = Q_H$. By using this in the heat conservation relation Eq. (9), we obtain

$$M = k_H h^{3/2} (\Delta T / T_G)^{3/2}. \qquad (25)$$

## 2.5 Steady-state regimes: melt-controlled and hydraulically-controlled exchange flows

The results presented above suggest that there can exist two different flow regimes in a fjord with basal ice-tongue melting: One where the exchange flow $Q$ is determined locally by the basal-melt processes on the glacier, and one where it is hydraulically-controlled. The two regimes have the following characteristics.

1. In the melt-controlled regime, the volume flow of the melt-water plume is smaller than the upper limit set by hydraulic control, i.e. $Q_P < Q_H$. Accordingly, the sill does not constrict the exchange flow, which is specified by Eq. (16) (or 21). The flow at the sill is subcritical (Pratt and Whitehead, 2007), and as result there is limited mixing between inflowing and outflowing waters. Essentially unmodified AW reaches the grounding line of the ice tongue (Fig. 1a), which implies that the thermal forcing is given by $\mathcal{T} = \mathcal{T}_A$. Since the freezing temperature ($T_f$) is set by the grounding-line depth (and $S_A$ which to a good approximation can be taken as constant here), the thermal forcing at a specific glacier is externally determined by the AW temperature ($T_A$).

2. In the hydraulically-controlled regime, the plume volume flow exceeds the hydraulic limit, i.e. $Q_P > Q_H$. The exchange flow is now determined by Eq. (23), and the flow at the sill crest is critical, and it accelerates down the landward slope of the sill (Pratt and Whitehead, 2007). Here the flow becomes supercritical, and inflowing AW mixes with colder outflowing water (Price and O'Neil Baringer, 1994; Pratt and Whitehead, 2007; Jakobsson et al., 2020; Schaffer et al., 2020). This lowers the temperature of the water reaching the grounding line (Fig. 1b), i.e. $\mathcal{T} < \mathcal{T}_A$. Importantly, this implies that the thermal forcing is no longer directly set by $\mathcal{T}_A$: $\mathcal{T}_A$ is the external forcing, but the local thermal forcing $\mathcal{T}$ is determined by dynamics in the fjord. The relationship between $\mathcal{T}$ and $\mathcal{T}_A$ can be expressed as

$$\mathcal{T} = R \cdot \mathcal{T}_A, \qquad (26)$$

where $R = R(\mathcal{T}_A, h)$ is a reduction factor which arises when the exchange flow is hydraulically controlled. Note that $R < 1$ in the hydraulic regime; in the melt-controlled regime $R = 1$.

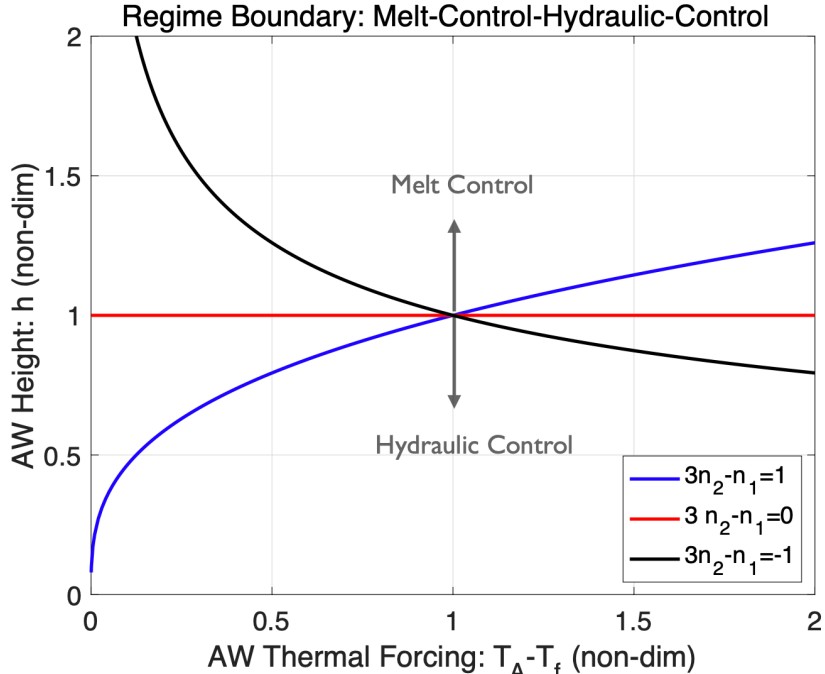

**Figure 4.** Regime boundaries between melt-controlled and hydraulically-controlled flows in the $\mathcal{T}_A$–$h$ plane (Eq. 27). The AW variables are non-dimensionalised by selecting an arbitrary scale for $\mathcal{T}_A$, and then define a non-dimensional $h$ that is one when the non-dimensional $\mathcal{T}_A$ is one; see Eq. (28). Hydraulically-controlled (melt-controlled) flows are found below (above) the lines, showing that the hydraulic regime is approached as $h$ is decreased. When $3n_2 - n_1$ is positive (negative), the transition height $h_L$ increases (decreases) with $\mathcal{T}_A$; and in the limiting case $3n_2 - n_1 = 0$, $h_L$ is independent of $\mathcal{T}_A = T_A - T_f$.

At the transition between the two flow regimes $\mathcal{T} = \mathcal{T}_A$ and $Q_P = Q_H$, which implies that the transports given by Eqs. (21) and (23) are equal. By using this and Eq. (18), which applies at the transition, we obtain after some rearrangements

$$\frac{\gamma_1^{1/3} k_H^{2/3} h}{\gamma_2 \mathcal{T}^{\frac{3n_2 - n_1}{3}}} = 1. \tag{27}$$

This represents the condition $Q_P/Q_H = 1$, and gives the relationship between $\mathcal{T}$ and $h$ at the regime transition. The height of the AW above the sill at the transition (say $h_L$) as a function of $\mathcal{T}_A$ is given by

$$h_L = k_H^{-2/3} \gamma_1^{-1/3} \gamma_2 \mathcal{T}_A^{\frac{3n_2 - n_1}{3}}. \tag{28}$$

---

[1]For simplicity, we assume a rectangular cross section, which implies that $W$ does not depend on $h$. Note that the lower-layer width $W$ may be smaller than the fjord width if the inflow is confined in a deeper channel crossing the sill, which is the case for Ryder; see Fig. 1 in Jakobsson et al. (2020).

When $h > h_L$, the flow is in the melt-controlled regime, and when $h < h_L$, the exchange flow becomes hydraulically controlled. Alternatively, Eq. (27) gives the thermal forcing at the regime transition as a function of $h$

$$\mathcal{T}_L = \left( \frac{k_H^2 h^3 \gamma_1}{\gamma_2^3} \right)^{\frac{1}{3n_2 - n_1}} . \tag{29}$$

Using this and Eq. (18), the temperature difference at the regime transition can be written as

$$\frac{\Delta T_L}{T_G} = \frac{\gamma_1 \mathcal{T}_L^{n_1 - n_2}}{\gamma_2} . \tag{30}$$

If $n_1 - n_2 = 1$, then $\Delta T_L = \sigma \mathcal{T}_L$; see Eq. (20).

Lowering the AW height always brings the flow towards the hydraulic regime. If $h$ is fixed then $\mathcal{T}_L$ is also fixed, which implies that changes of the AW temperature can cause a transition between the melt-controlled and the hydraulically-controlled regimes. As illustrated in Fig. 4, the nature of the transition depends on the value of the exponent $3n_2 - n_1$ in Eq. (27). If $3n_2 - n_1 > 0$ increasing $\mathcal{T}_A$ will bring the flow towards the hydraulically-controlled regime. On the other hand if $3n_2 - n_1 < 0$, decreasing $\mathcal{T}_A$ will brings the flow towards the hydraulically-controlled regime. This behaviour follows from the fact that $Q_P \propto \Delta T^{\frac{n_2}{n_1 - n_2}}$ and $Q_H \propto \Delta T^{1/2}$. In the case $3n_2 - n_1 = 0$, the regime transition height $h_L$ is independent of $\mathcal{T}_A$. This is because $Q_P$ and $Q_H$ then have the same dependence on $\Delta T$.

Figure 5 illustrates the relation between exchange flow and temperature difference (Eqs. 21,23) for the case $n_1 = 2$ and $n_2 = 1$, which implies that $Q_P \propto \Delta T$. Here, the flow is in the melt-controlled regime if $\Delta T < \Delta T_L$, or from Eq. (18) equivalently if $\mathcal{T}_A < \mathcal{T}_L$. By increasing $\Delta T$, the flow increases and is shifted towards and into hydraulic control.

In the case where $3n_2 - n_1 < 0$, hydraulic control – for a fixed $h$ – occurs for weak thermal forcing and exchange flow. In a real fjord, additional exchange flows driven by winds and tides may be larger than a model predicted weak hydraulic flow (Jackson and Straneo, 2016). This can prevent establishment of hydraulic control and in effect yield an exchange flow in the melt-controlled regime. We will briefly discuss this case in section 3.2.5 .

It is possible that transitions between melt- and hydraulically-controlled regimes can also be caused by seasonal variations in subglacial discharge, even if the AW features remain unchanged. The reason is that the basal melt $M$ increases with the subglacial discharge (Jenkins, 2011; Xu et al., 2013), which has pronounced seasonal cycle that tracks the surface melt on the glaciers (e.g. Truffer and Motyka, 2016; Cai et al., 2017; Slater and Straneo, 2022). Thus, it is possible that in some fjords the exchange circulation can be stronger and hydraulically-controlled in summer when the subglacial discharge peaks, and weaker and melt-controlled in winter. The details of such seasonal regime transitions will depend, among other things, on how the ratio between exchange flow and melt ($M/Q$) depends on the subglacial discharge. However, we will not pursue this topic further.

## 3 The dynamics in the hydraulic regime

In the hydraulic regime, the volume flow of the melt plume is predicted to exceed the exchange flow at the sill (Fig. 5). This would cause an imbalance in production and export of glacially-modified waters in the fjord, preventing a steady state to be established. Therefore, changes of the flow are expected when hydraulic control is established.

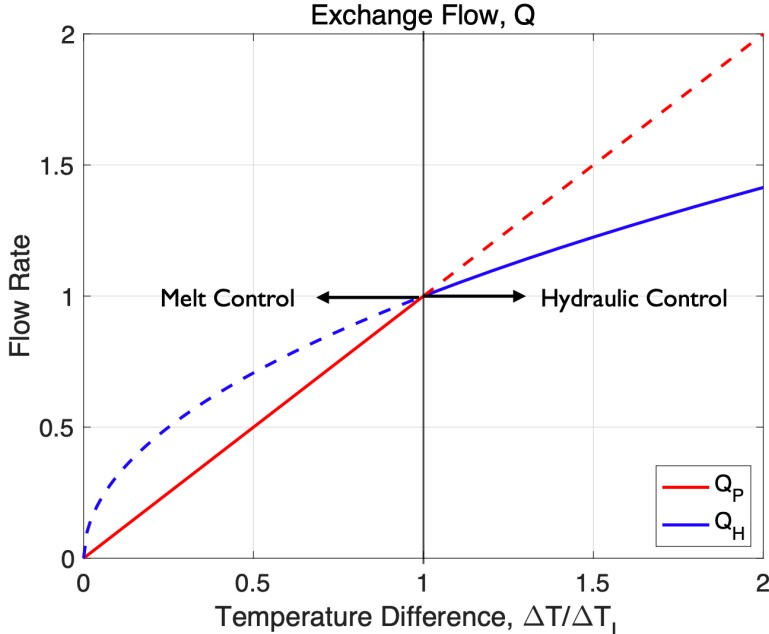

**Figure 5.** The hydraulically-controlled flow $Q_H$ and the plume volume flow $Q_P$ as functions of the layer temperature difference $\Delta T$, for a fixed AW height ($h$). The case $n_1 = 2$ and $n_2 = 1$ is shown, and $\Delta T$ is normalised by $\Delta T_L$ (Eq. 30). The flow $Q_H$ (blue lines) is proportional to $\Delta T^{1/2}$ (Eq. 23), and $Q_P$ (red lines) is proportional to $\Delta T$ (Eq. 21). The solid lines show the actual exchange flow, which is set by the lower value of the two flows.

To examine some of the new features emerging when the flow becomes hydraulically controlled, it is instructive to consider a thought experiment in which the sill height is suddenly increased and hydraulic control is established. Initially, the production of glacially-modified water will be larger than the exchange flow across the sill. As a result, the layer of glacial water inside the sill will thicken and possibly extend below the sill crest (Fig. 1). This has two important consequences for the basal melt. First, the inflowing AW will entrain glacial water, causing the temperature of water reaching the grounding line to decrease. 320 Second, the melt-water plume will rise partly through ambient waters that are colder and lighter than the displaced AW. This reduces the buoyancy and speed of the plume, which now also will entrain colder water. Theses changes of the stratification in the ice cavity act to reduce the basal melt. Thus, we expect that the temperature and salinity distributions inside the sill evolve such that a new steady state, compatible with the hydraulically-constrained exchange flow, is established.

The reasoning above suggests that, in the hydraulic regime, the interface height of (pure or modified) AW is no longer the 325 same on the seaward and landward side of the sill. Thus additional variables, such as a fjord interface height, may be needed to model flow- and melt-features in the hydraulic regime. However, we will not introduce additional model variables. Instead, we consider two idealised scenarios for how the interplay between melt dynamics and hydraulic control can determine the steady state flow. In these scenarios, the features of the fjord stratification (i.e. interface height and layer difference of temperature and

salinity) can be viewed as hidden model variables that influence the flow. In Scenario 1, we implicitly assume that no glacially-modified water is entrained into the inflowing AW near the sill. This is an extreme and less likely scenario as observations and modelling show that entrainment generally occur (Schaffer et al., 2020; Jakobsson et al., 2020; Hager et al., 2022; Bao and Moffat, 2023). In Scenario 2, on the other hand, entrainment plays a key role for closing volume budget in the ice cavity.

## 3.1 Scenario 1: Hydraulically constrained plume volume flow

### 3.1.1 Physical assumptions

Here, we assume that the steady-state flow and stratification inside the sill adjust such that:

1. The plume volume flow (Eq. 16) and the exchange flow (Eq. 23) are equal, which implies that $Q_P(\mathcal{T}) = Q_H(h, \Delta T)$.

2. The relationship between basal melt $M$ and thermal forcing (Eq. 15) still applies and equals the formula for $M$ in the hydraulically-controlled regime (Eq. 25). This gives a relationship of the form $\mathcal{T} = \mathcal{T}(h, \Delta T)$.

From these assumptions, we obtain the following expressions for the thermal forcing and temperature difference

$$\mathcal{T} = (k_H^2 h^3 \gamma_1 \gamma_2^{-3})^{\frac{1}{3n_2 - n_1}}, \tag{31}$$

$$\Delta T / T_G = \gamma_1 \gamma_2^{-1} (k_H^2 h^3 \gamma_1 \gamma_2^{-3})^{\frac{n_1 - n_2}{3n_2 - n_1}}. \tag{32}$$

Note that $\mathcal{T}$ and $\Delta T$ depend both on the features of the melt representation and the hydraulic constant $k_H$. In the reference case ($n_1 = 2$ and $n_2 = 1$) the exponents in the expressions above simplify, and the hydraulic exchange flow (Eq. 23) becomes

$$Q_H = k_H^2 h^3 \gamma_1 \gamma_2^{-2}. \tag{33}$$

Notably, the flow is independent of the AW temperature $T_A$: The strength of the hydraulically-controlled flow is determined by $h$ and the parameters $\gamma_1$, $\gamma_2$, and $k_H$, which control $\Delta T$ that is proportional to layer density difference.

### 3.1.2 Dynamical features

Figure 6 shows the dependence of the thermal forcing and the reduction factor on the AW forcing ($\mathcal{T}_A$ and $h$) in the case $n_1$ and $n_2$. The flow is in the hydraulic regime when $\mathcal{T}_A > \mathcal{T}_L$ and $h < h_L$, and the opposite applies in the melt-controlled regime. In the melt-controlled regime, $\mathcal{T}$ is equal to $\mathcal{T}_A$, and is independent of $h$. An increase in $\mathcal{T}_A$ or a decrease in $h$ brings the flow towards the hydraulically-controlled regime. In this regime, the flow features become (in this scenario) independent of the AW temperature. As a result, the $R$ factor decreases with increasing $\mathcal{T}_A$ at a fixed $h$. This provides a negative feedback on the basal melt. The flow features are sensitive to changes of $h$: when $n_1 = 2$ and $n_2 = 1$, one finds that $\mathcal{T} \propto h^3$ and $M \propto h^6$. Accordingly, the basal melt drops sharply with decreasing AW height.

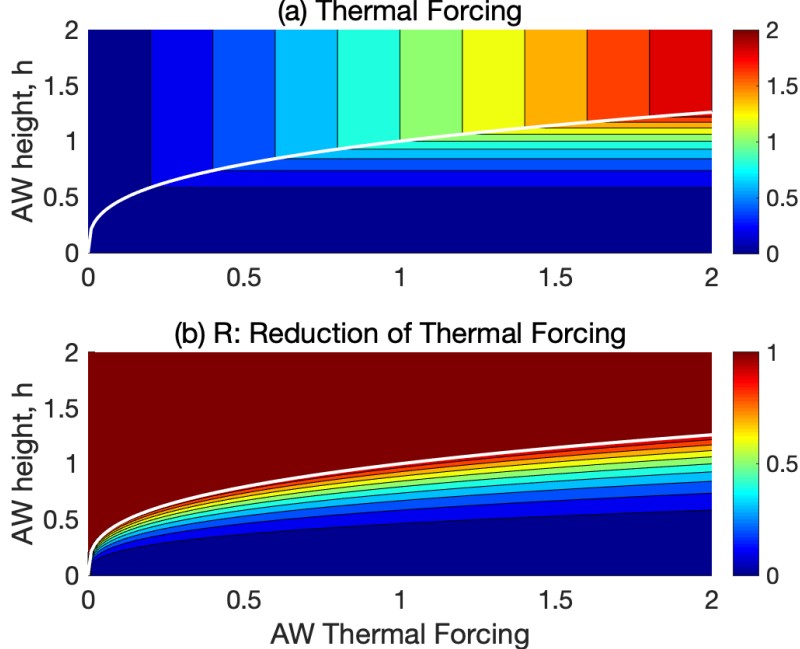

**Figure 6.** The dependence of the thermal ice-cavity forcing $\mathcal{T}$ (a) and the reduction factor $R$ (b) on the AW thermal forcing ($\mathcal{T}_A$) and height ($h$) in scenario 1; see section 3.1. Here, $\mathcal{T} = T_C - T_f$ and $R = \mathcal{T}/\mathcal{T}_A$ (Eqs. 13, 26), implying that $R = 1$ in the melt-controlled regime. The non-dimensional variables are selected such that $\mathcal{T} = 1$ when $\mathcal{T}_A = 1$ and $h = 1$. The white line shows the boundary between the melt-controlled (above the line) and hydraulically-controlled regime (below the line). The case $n_1 = 2$ and $n_2 = 1$ is shown.

In this hypothetical scenario, one should view $\mathcal{T}$ as an effective thermal forcing, which can have an implicit dependence on features such as the fjord stratification, rather than the actual thermal forcing near the grounding line (i.e., $T_C - T_f$). We expect that a a sudden increase of the inflowing AW temperature at the sill would initially cause warmer water to reach the grounding line and increase $Q_P$ and $M$. However, adjustments of the temperature and stratification in the ice cavity are assumed to re-establish a state with the $T_A$ independent melt rate determined by Eqs. (31) and (15). Since $\Delta T = T_A - T$ is constant, the outflow temperature $T$ mirrors $T_A$.

## 3.2 Scenario 2: Unconstrained plume volume flow

### 3.2.1 Physical assumptions

We will now consider another hypothetical scenario in which the plume volume transport is determined by the thermal forcing via Eq. (16) and is not set directly by the hydraulic constraints. This implies that $Q_P > Q_H$, and additional physical processes need to be invoked to balance the production and export over the sill of glacially-modified water. Entrainment of some glacially-modified water into the inflowing AW, will help to achieve this balance: The entrained glacially-modified water is, in effect,

re-circulating in the basin inside the sill. This can allow a steady state to develop even if $Q_P$ exceeds $Q_H$. Specifically, in this scenario, we assume that:

1. The excess volume flow in the plume is supplied by entrainment of glacially-modified water ($Q_E$) into the inflow of AW on the landward side of the sill:

$$Q_P = Q_E + Q_H. \tag{34}$$

By introducing the entrainment fraction (Price and O'Neil Baringer, 1994)

$$\Phi \stackrel{\text{def}}{=} \frac{Q_E}{Q_H + Q_E} = 1 - \frac{Q_H}{Q_P}, \tag{35}$$

we can relate the plume volume flow and the exchange flow as

$$Q_P = \frac{Q_H}{1 - \Phi}. \tag{36}$$

The parameter $\Phi$ ranges from 0 (no entrainment) to 1 (in the limit of strong entrainment). We note that $\Phi$ is essentially equal to the re-flux factor used by Hager et al. (2022).

2. The relationship between basal melt $M$ and thermal forcing (Eq. 15) still applies and equals the formula for $M$ in the hydraulically-controlled regime (Eq. 25).

The second assumption yields the following relationship

$$\frac{\Delta T}{T_G} = \left( \frac{\gamma_1 \mathcal{T}^{n_1}}{k_H h^{3/2}} \right)^{2/3}. \tag{37}$$

By using this result in Eq. (23), we obtain

$$Q_H = \left( k_H^2 h^3 \gamma_1 \mathcal{T}^{n_1} \right)^{1/3}. \tag{38}$$

These formulas, which depend on features of the basal melt as well as $h$ and $k_H$, specify the flow dependence on the thermal forcing in this hydraulic-regime scenario. However, $\mathcal{T}$ is a function of the AW forcing $\mathcal{T}_A$ and $h$ that remains to be determined. This is obtained by considering how the grounding-line temperature is affected by entrainment of glacially-modified waters as outlined below.

The entrainment is controlled by local conditions on the landward side of the sill, where the denser inflowing AW accelerates down the sill slope (Price and O'Neil Baringer, 1994; Pratt and Whitehead, 2007). However, we assume for simplicity that $Q_E$ adjusts to satisfy Eq. (34). By using Eqs. (16,38), we can after some manipulations express the entrainment rate $\Phi$ as

$$\Phi = 1 - Z, \tag{39}$$

where

$$Z \stackrel{\text{def}}{=} \left( \frac{\gamma_1^{1/3} k_H^{2/3} h}{\gamma_2 \mathcal{T}^{\frac{3n_2 - n_1}{3}}} \right). \tag{40}$$

Here $\Phi$ is given by Eq. (39) when $Z \leq 1$, and when $Z > 1$ $\Phi = 0$. Note that $Z = 1$ yields the condition (Eq. 27) that defines the flow-regime transition.

Next, we consider the relationship between the AW ($S_A$, $T_A$) and the water properties in the ice cavity ($S_C$, $T_C$), which are affected by the entrainment: Conservation of heat in the cavity (see Fig. 1) yields

$$T_A Q_H + T Q_E = T_C(Q_H + Q_E). \tag{41}$$

By using the entrainment parameter $\Phi$ (Eq. (35), this yields the following expression for the temperature in the ice cavity

$$T_C = T_A - \Delta T \Phi. \tag{42}$$

Conservation of salt yields an analogous expression for $S_C$. Finally by using Eqs. (13,42), we obtain

$$\mathcal{T} = \mathcal{T}_A - \Delta T \Phi. \tag{43}$$

This relation and Eqs. (37,39) determine the functional relationship $\mathcal{T} = \mathcal{T}(\mathcal{T}_A, h)$ in this scenario. Note that since $\Delta T$ and $\Phi$ are specified as functions of $\mathcal{T}$ and $h$, Eq. (43) also yields the function $\mathcal{T}_A(\mathcal{T}, h) = \mathcal{T} + \Delta T \Phi$, which is algebraically easier to use when constructing graphical solutions. This is because the function $\mathcal{T}(\mathcal{T}_A, h)$ cannot, generally, be obtained on a closed analytical form; see appendix B.

### 3.2.2 Dynamical features: general aspects

Figure 7 shows, for scenario 2, the dependence of the thermal forcing and the reduction factor on the AW forcing ($\mathcal{T}_A$ and $h$). Again, the case $n_1 = 2$ and $n_2 = 1$ is illustrated, which is qualitatively representative for melt representations satisfying $3n_2 > n_1$. Qualitatively, the behaviour is similar to that of scenario 1 (Fig. 6). A difference is that the thermal forcing now increases with $\mathcal{T}_A$ in the hydraulic regime. However, the rate of increase is weaker than linear ($\frac{d\mathcal{T}}{d\mathcal{T}_A} < 1$).

For the hydraulically-controlled flow, entrainment of colder glacially-modified waters into the inflowing water lowers the ice cavity temperature relative to $T_A$. This decreases the thermal forcing and thereby the melt rate: $\mathcal{T}$ decreases with decreasing $h$ and increases more slowly with $\mathcal{T}_A$ than in the melt-controlled regime. Figure 7b shows the reduction factor $R$ (Eq. 26). By definition $R = 1$ in the melt-controlled regime. In the hydraulic regime, $R < 1$, and the sensitivity of the thermal forcing to changes in $\mathcal{T}_A$ is reduced: the isolines of constant $R$ becomes shallower with increasing $\mathcal{T}_A$. In the present scenario 2, the flow response depends also on the parameter $\sigma$, a non-dimensional measure of the outflow temperature $T$ at the regime transition, which will be discussed below.

### 3.2.3 Dynamical features: dependence on AW height

Here, we examine the flow and melt response to changes in the AW height $h$ for a fixed $\mathcal{T}_A$, i.e. moving vertically in Fig. 7. Specifically, we consider how the parameter $\sigma$ affects the response. Recall that $\sigma$ is a non-dimensional measure of the outflow temperature $T$ in melt-controlled regime where $\Delta T / \mathcal{T}_A = \sigma$; see Eq. (20). We will consider the whole range of possible $\sigma$ values ($0 \leq \sigma \leq 1$), but our observationally-based estimates indicate that $\sigma$ is about 0.1 (Table 2).

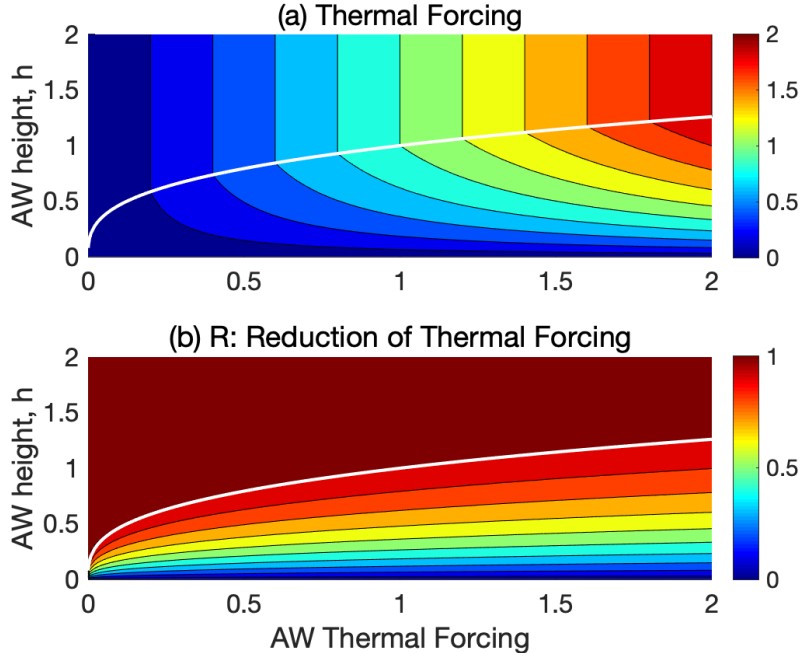

**Figure 7.** The dependence of the thermal ice-cavity forcing $\mathcal{T}$ (a) and the reduction factor $R$ (b) on the AW thermal forcing ($\mathcal{T}_A$) and height ($h$) in scenario 2; see section 3.2. Here, $\mathcal{T} = T_C - T_f$ and $R = \mathcal{T}/\mathcal{T}_A$ (Eqs. 13, 26), implying that $R = 1$ in the melt-controlled regime. The white line shows the boundary between the melt-controlled (above the line) and hydraulically-controlled regime (below the line). The non-dimensional variables are defined as in Fig. 6. The case $n_1 = 2$ and $n_2 = 1$ is shown for $\sigma = 0.5$; see Eq. (20) and the text.

Figure 8 illustrates how $Q$, entrainment fraction $\Phi$, $\Delta T$, $\mathcal{T}$ and $M$ vary with the AW height $h$. (In the figures, we have normalised $Q$, $\mathcal{T}$ and $M$ to be unity at the regime transition; but $\Delta T$ is normalised equal to $\sigma$.) If $h$ is decreased, either by an increase in the sill height or by lowering the upper boundary of the AW, the flow is unchanged until $h = h_L$, the point at which the flow becomes hydraulically controlled. By further reducing $h$, $Q$, $\mathcal{T}$ and $M$ decline but $\Delta T$ grows (implying decreasing $T$). In the limiting case $\sigma = 1$, where $\Delta T$ is constant, $Q$ is proportional to $h^{3/2}$; see Eq. (23). When $\sigma < 1$, $Q$ falls less steeply

with $h$ because the layer density difference (proportional to $\Delta T$) increases with decreasing $h$.

In the hydraulic regime, outflowing glacially-modified water is entrained into inflowing AW, thereby reducing $\mathcal{T}$ (Fig. 8c). This effect is most pronounced for larger values of $\sigma$, which correspond to colder outflow temperatures of the glacially-modified water. In the limiting case $\sigma = 1$, $M \propto h^{\frac{3}{2}}$ because $\Delta T = \mathcal{T}_A$ is constant; see Eq. (25). The dependence of $M$ on $h$ in this limiting case describes the response for all values of $\sigma$ when $h/h_L$ becomes small. Here, $\Delta T/\mathcal{T}_A$ approaches its maximum

value of one. By using that $\Delta T \approx \mathcal{T}_A$ in Eq. (25), the melt rate becomes

$$M \approx k_H (h\mathcal{T}_A/T_G)^{3/2}. \tag{44}$$

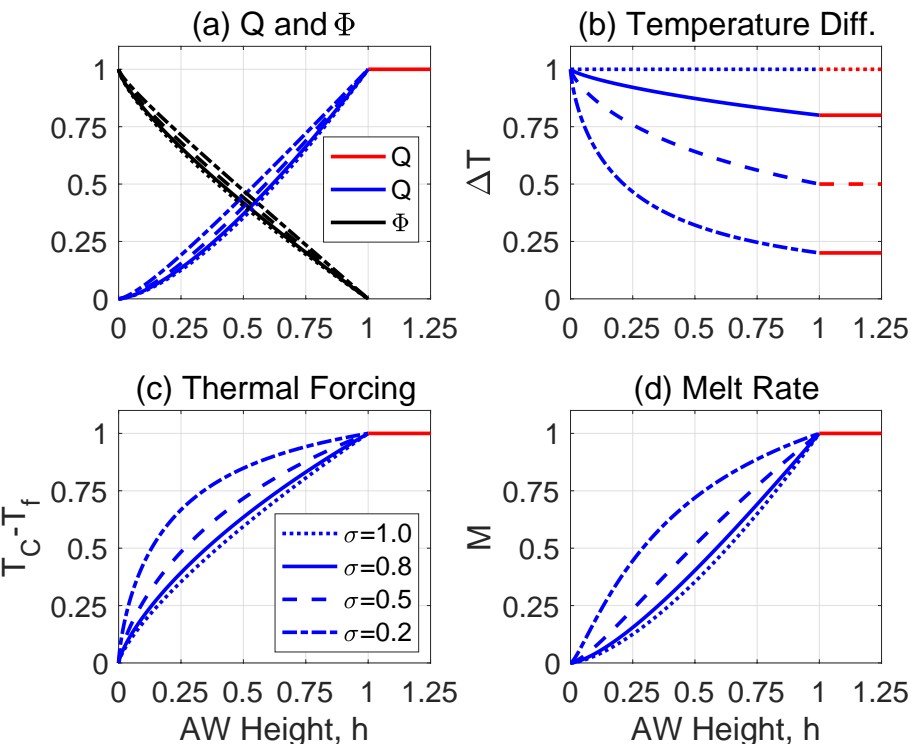

**Figure 8.** The flow dependence on the AW height $h$ for a fixed value of $\mathcal{T}_A$. The case $n_1 = 2$ and $n_2 = 1$ is shown for different values of the non-dimensional parameter $\sigma$ (Eq. 20); dotted, solid, dashed, and dash-dotted lines show results for $\sigma = 1.0$, $\sigma = 0.8$, $\sigma = 0.5$, and $\sigma = 0.2$, respectively. All variables are non-dimensional: the AW height $h/h_L$ (Eq. 28) is smaller (greater) than one in the hydraulic (melt-controlled) regime. (a) Exchange flow $Q$ (blue and red lines) and entrainment fraction $\Phi$ (black lines), and (b) the temperature difference $\Delta T$. (c) and (d) shows the thermal forcing ($\mathcal{T}/\mathcal{T}_A$) and melt rate. The melt rates are normalised to be unity in melt-controlled regime; dimensional melt rates are proportional to $\sigma^{3/2}$ (see the text).

Notably, the melt rate is independent of the features of the melt representation in this limit. From the melt rate formula (15), it follows that the thermal forcing is approximately given by

$$\mathcal{T} \approx \left( \frac{k_H h^{3/2}}{\gamma_1} \right)^{\frac{1}{n_1}} \left( \frac{\mathcal{T}_A}{T_G} \right)^{\frac{3}{2n_1}}. \tag{45}$$

Thus, the thermal forcing still depends on the melt parameterisation, but in such a way that the melt itself only depends on $k_H$, $h$ and $\mathcal{T}_A$.

    To summarise, in scenario 2 hydraulic control constrains the exchange flow and induces entrainment, which acts to lower the water temperature at the grounding line relative to $T_A$. Notably, when $h/h_L$ becomes sufficiently small, the flow enters a regime where the exchange flow and the basal melt become independent of the physical processes near the ice–ocean interface 445   that govern the local melt rates. A partly analogous situation is an over-mixed estuary, where hydraulic control at a sill or a

fjord mouth sets the exchange flow rate and vertical salinity difference independently of the nature of the mixing processes in the estuary (Stommel and Farmer, 1953; Timmermans, 1998).

### 3.2.4 Dynamical features: dependence on AW temperature

Next, we consider how the melt and flow features depend on the AW temperature when $h$ is fixed. Figure 9 a and b show how the exchange flow $Q$ (normalised to be unity at the regime transition), temperature difference $\Delta T$, the entrainment fraction $\Phi$ varies with the $\mathcal{T}_A$. In the melt-controlled regime, $Q$ and $\Delta T$ depend linearly on $\mathcal{T}_A$ (when $n_1 = 2$ and $n_2 = 1$), and a larger $\sigma$ is associated with a larger $\Delta T$. If $\mathcal{T}_A$ is increased, the flow enters the hydraulically-controlled regime, where $Q$ is proportional to $\Delta T^{1/2}$. This constrains the exchange flow, and increasing entrainment lowers the temperature at the grounding line. In response, the outflow temperature decreases, which causes $\Delta T$ to increase with $\mathcal{T}_A$ at rate that is slightly higher than linear. In the limiting case $\sigma = 1$, $\Delta T = \mathcal{T}_A$ in both regimes and hence $Q$ is proportional to $\mathcal{T}_A^{1/2}$ in the hydraulic regime. The entrainment rate depends only weakly on $\sigma$ and increases relatively slowly with $\mathcal{T}_A$.

Figure 9 c and d show the dependence of the thermal forcing and basal melt on $\mathcal{T}_A$, which both are normalised to be unity at the regime transition. In the hydraulic regime, $\mathcal{T}$ and $M$ are for a given $\mathcal{T}_A$ lower than they would have been in a melt-controlled regime. Since $M \propto \Delta T Q$, the hydraulic-regime melt rate is proportional to $\mathcal{T}_A^{3/2}$ when $\sigma = 1$. Note that in Fig. 9, the melt rates have been normalised to be unity at the regime transition for visual clarity. The actual melt rates are proportional to $\sigma^{3/2}$, implying that $\sigma = 1$ corresponds to the highest melt rate of a hydraulically-controlled exchange flow for a given $\mathcal{T}_A$.

### 3.2.5 Dynamical features: dependence on the melt parameterisation exponents $n_1$ and $n_2$

So far we have considered the case $3n_2 - n_1 > 0$, where increasing $\mathcal{T}_A$ brings the flow towards the hydraulic regime (Fig. 4). The large 79°, Petermann and Ryder ice tongues should be descried by this case. However, some qualitatively different flow features emerge if $3n_2 - n_1 < 0$, and this case may be relevant for tidewater glaciers with high glacial discharge: theoretical considerations sugget that the exponents $n_1 = 1$ and $n_2 = 0$ describe the melt processes in this limit (Jenkins, 2011; Straneo and Cenedese, 2015). Therefore, we consider briefly two cases for which $3n_2 - n_1 \leq 0$ in the context of scenario 2.

Figure 10 shows flow features for the cases $3n_2 - n_1 = 0$ and $3n_2 - n_1 = -1$. In the former case, the flow has the same dependence on $\mathcal{T}$ and $\Delta T$ in both regimes; see Eqs. (16,38). As a result, the boundary between the flow regimes depends only on $h$. Further the $R$ factor, the suppression of the thermal forcing due to hydraulic control, is independent of $\mathcal{T}_A$.

Also in the case where $3n_2 - n_1 < 0$, a hydraulically-controlled exchange flow suppresses the thermal forcing and basal melt. However, here the $R$ factor – for a fixed $h$ – increases with increasing $\mathcal{T}_A$. Thus, as the AW temperature increases the hydraulic suppression of the melt decreases. The reason is that the hydraulically-determined upper bound on the exchange flow $Q_H$ (Eq. 38) now increases faster with the thermal forcing than the plume volume transport $Q_P$ (Eq. 16). Figure 10 (b) and (d) illustrate the case $n_1 = 1$ and $n_2 = 0$ where $Q_P$ is independent of the thermal forcing, but capture the qualitative features for the general case $3n_2 - n_1 < 0$.

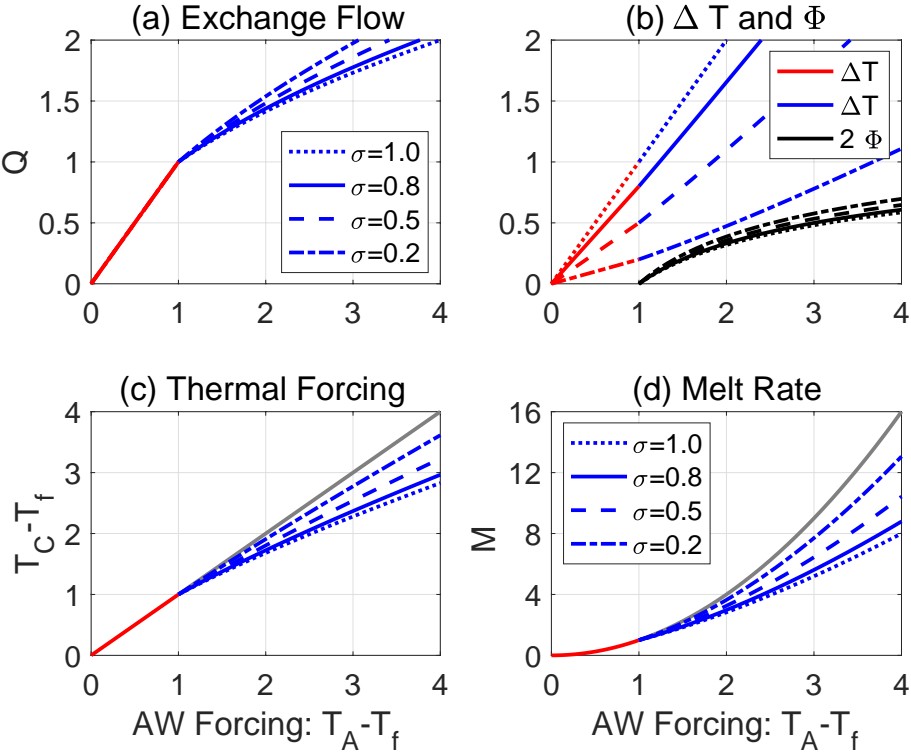

**Figure 9.** The flow dependence on the AW thermal forcing $\mathcal{T}_A$ for a fixed value of $h$. The case $n_1 = 2$ and $n_2 = 1$ is shown for different values of the non-dimensional parameter $\sigma$ (Eq. 20); solid, dashed, and dash-dotted lines show results for $\sigma = 1.0$, $\sigma = 0.8$, $\sigma = 0.5$, and $\sigma = 0.2$, respectively. All variables are non-dimensional: the AW thermal forcing $\mathcal{T}_A/\mathcal{T}_L$ (Eq. 29) is greater (smaller) than one in the hydraulic (melt-controlled) regime. (a) Exchange flow Q and (b) temperature difference $\Delta T$ (red and blue lines), and entrainment fraction $\Phi$ (black lines); note that for clarity $2\Phi$ is graphed. (c) Thermal forcing $\mathcal{T}$ and (d) and melt rate $M$; the grey lines show $\mathcal{T}$ and $M$ in the melt-controlled regime extrapolated into the hydraulic regime. Note that $M$ have been normalised to be unity at the regime transition; dimensional melt rates are proportional to $\sigma^{3/2}$ (see the text).

## 3.3 Applications to Ryder, 79°N, and Petermann glaciers

To examine some concrete aspects of the model, we will now apply it in a qualitative way to the ice tongues of Ryder, 79°N, and Petermann glaciers. We consider scenario 2, assuming $n_1 = 2$ and $n_2 = 1$, and try to crudely estimate model parameters charactering the melt–flow dynamics. We use observations of flow- and melt-rates ($Q$ and $M$) and hydrography from the three glaciers reported in the literature (Johnson et al., 2011; Wilson et al., 2017; Jakobsson et al., 2020; Schaffer et al., 2020). We recall that observations show that the sill exchange flows of Ryder and 79°N glaciers are hydraulically controlled, but that the exchange flow over the relatively deep and wide sill in Peterman Fjord is not. Note that there are uncertainties in the observations and in model assumptions, and the present exercise is primarily an illustration of how the model can be applied.

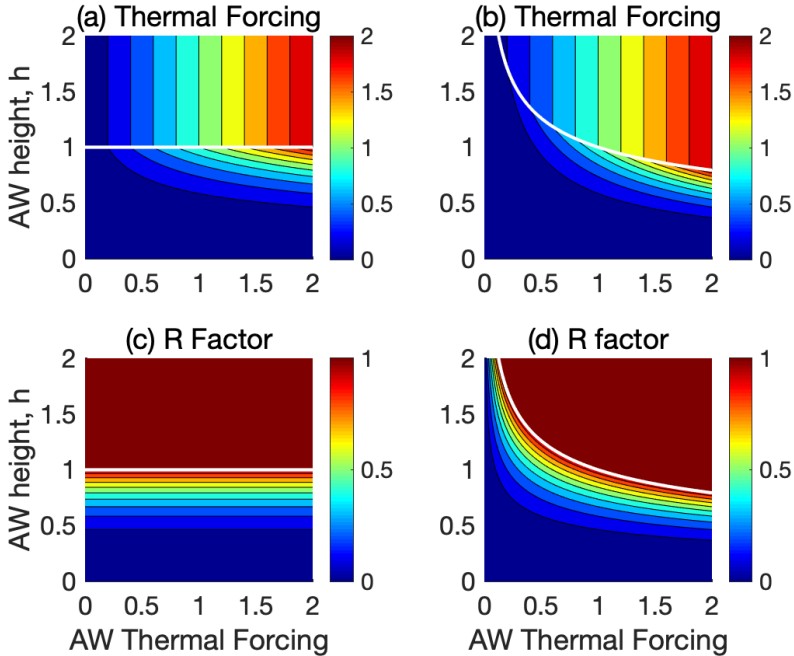

**Figure 10.** The dependence of the thermal ice-cavity forcing $\mathcal{T}$ (a,b) and the reduction factor $R$ (c,d) on the AW thermal forcing ($\mathcal{T}_A$) and height ($h$) in scenario 2; see section 3.2. The non-dimensional variables are defined as in Fig. 6, and $\sigma = 0.5$. The white line shows the boundary between the melt-controlled (above the line) and hydraulically-controlled regime (below the line). (a) and (c) show the case $n_1 = 1.5$ and $n_2 = 0.5$, for which $3n_2 - n_1 = 0$; and (b) and (d) show the case $n_1 = 1$ and $n_2 = 0$, for which $3n_2 - n_1 = -1$. The latter case may be relevant of tidewater glaciers, but for large ice tongues the case $3n_2 - n_1 > 0$ shown in Fig. 7 is more likely.

Guided by the model physics, we estimate model variables and parameters as follows: The hydrographic observations give $T_C$ as the near-bottom temperatures inside the sill, and $T_A$ as the sill-depth temperature outside the the sill. Note that the outside the sills, the temperatures are nearly constant below the sill depths; see Fig. 3. The outflow model temperature $T$, which represents a flow-weighted mean over of an outflow distributed vertically over a range of temperatures, is less straightforward to determine from hydrography. Here, we use the relation $M/Q = \Delta T/T_G = \Delta S/S_A$ [which follows from Eqs. (3,9)] to find
values of $T$ and $S$ that roughly satisfy these conditions, and at the same time characterise outflowing water. This allows $\Phi$ to be determined from Eq. (42).

     The model parameters $\gamma_1$, $\gamma_2$, and $k_H$ are estimated as follows: From Eq. (15), we obtain $\gamma_1 \approx M/\mathcal{T}^2$. By using Eqs. (15,16,36), we obtain $\gamma_2 \approx Q/[\mathcal{T}(1-\Phi)]$. This provides the estimate

$$\sigma = \frac{T_G \gamma_1}{\gamma_2} \approx \frac{T_G M}{\mathcal{T} Q}(1 - \Phi). \tag{46}$$

By using Eqs. (23,25), we obtain $k_H h^{3/2} \approx Q^{3/2}/M^{1/2}$; and an estimate of $h$ then gives $k_H$. Note that $k_H$ can also be determined from a knowledge of the cross-sectional sill width $W$.

**Table 2.** Observational features and estimated model parameters in scenario 2 (section 3.2) for Ryder, 79°N, and Petermann glaciers. See the text for details. In the model fit, it is assumed that $n_1 = 2$ and $n_2 = 1$. Note that the AW temperature $T_A$ is the measured temperature on the seaward side of the sill closest to the ice tongue.

| | Ryder Glacier | 79°N Glacier | Petermann |
|---|---|---|---|
| $A$ area (km$^2$) | 300 | 1700 | 1000 |
| $M$ (m$^3$ s$^{-1}$) | 60 | 600 | 300 |
| $Q$ ($10^3$ m$^3$ s$^{-1}$) | 10 | 46 | 50 |
| $M/Q$ (%) | 0.6 | 1.3 | 0.6 |
| $T_A, T_C$ (°C) | 0.3, 0.2 | 1.8, 1.2 | 0.3, 0.3 |
| $\mathcal{T}_A, \mathcal{T}$ (°C) | 2.8, 2.7 | 4.2, 3.6 | 2.8, 2.8 |
| $R = \mathcal{T}/\mathcal{T}_A$ | 0.96 | 0.86 | 1 |
| $\Delta T$ (°C) | 0.5 | 1.0 | 0.5 |
| $\Delta S$ (g/kg) | 0.2 | 0.5 | 0.2 |
| $\Phi$ | 0.3 | 0.6 | 0.0 |
| $\gamma_1$ (m$^3$ s$^{-1}$ °C$^{-2}$) | 8 | 46 | 40 |
| $\gamma_2$ (m$^3$ s$^{-1}$ °C$^{-1}$) | $5 \cdot 10^3$ | $3 \cdot 10^4$ | $2 \cdot 10^4$ |
| $k_H h^{3/2}$ (m$^3$ s$^{-1}$) | $1.3 \cdot 10^5$ | $4 \cdot 10^5$ | |
| $h_L$ (m), $h/h_L$ | 100, 0.7 | 220, 0.4 | 40, 3.4 |
| $\sigma$ | 0.1 | 0.1 | 0.2 |

The estimates of $Q$ from Ryder and Petermann are more uncertain than the ones from 79°N reported by Schaffer et al. (2020), which are based on a one-year moored time series of velocity. The Ryder estimate of $Q$ is based on a single instantaneous current measurement on the inner sill in Sherard Osborn Fjord (Jakobsson et al., 2020). Our Petermann estimate of $Q$ is based on hydrography and geostrophic velocities presented by Johnson et al. (2011): using their Fig. 7, we estimate the outflow of glacially modified water (in the depth rage from 150 to 250 m) to be on the order of $50 \cdot 10^3$ m$^3$ s$^{-1}$.

Table 2 summarises observational features and estimated model parameters. Notably, the estimates of $\sigma$, around 0.1–0.2, are similar for the three glaciers. This reflects that the cooling due to ice melt of the melt-water plume is small compared to the upper limit ($\sigma = 1$), in which the outflow temperature ($T$) approaches the freezing point. For Ryder and 79°N glaciers, which have hydraulically-controlled exchange flows, the estimated $h/h_L$ is 0.7 and 0.4, respectively. The lower value of $h/h_L$ at 79°N Glacier implies a higher sensitivity of the melt to changes of $h/h_L$; see Fig. 8. Further, the reduction factors ($R = \mathcal{T}/\mathcal{T}_A$), which are directly inferred from the hydrography, are only slightly below unity: $R$ is 0.96 and 0.86 at Ryder and 79°N, respectively. If as assumed here $M \propto \mathcal{T}^2$, this implies that, relative the situation where unmodified AW reaches the grounding line, the melt rates are reduced by about 10 and 30 % at Ryder and 79°N, respectively. Taken together, this suggests that hydraulic control and associated entrainment reduce the basal melt on both ice tongues, but that currently this effect is more pronounced at 79°N Glacier.

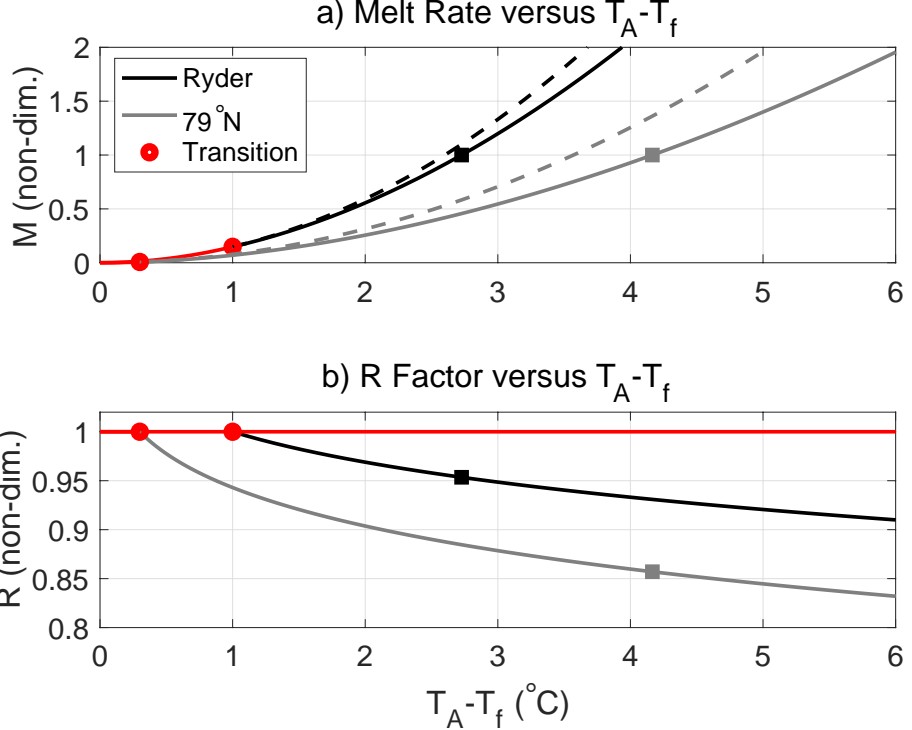

**Figure 11.** Model-based estimates of non-dimensional melt rate $M$ (a) and $R$ factor (b), for observed values of $h$, as a function of the AW thermal forcing $\mathcal{T}_A$ for Ryder (black lines) and 79°N (grey lines) glaciers. Red lines shows the melt-controlled regime, and dashed lines in (a) shows $M$ if AW would reach the grounding line ($\mathcal{T} = \mathcal{T}_A$). The estimate is based on the model scenario 2 (see 3.2 and with $n_1 = 2$ and $n_2 = 1$). The melt rates are normalised to be unity for the present observed values; see table 2 and section 3.3. The squares mark the observed values of $\mathcal{T}_A$, and the red dots mark the model-predicted transition between the melt-controlled and the hydraulically-controlled regimes.

Figure 11 shows model-predicted melt rates $M$ and $R$ factors as a function of $\mathcal{T}_A$ for Ryder and 79°N glaciers when $h$ is kept at observed values. The figure also shows the melt rates that would occur if unmodified AW reached the grounding lines: the melt rates in the hydraulically-controlled regime are lower and their dependence on $\mathcal{T}_A$ are weaker. The model results show

that a 1 °C increase of $\mathcal{T}_A$ from present values increases $M$ with about 75 % and 50 % at Ryder and 79°N, respectively. The basal melt is more sensitive to the same change in $T_A$ at Ryder than at 79°N simply because $\mathcal{T}_A$ is larger at 79°N. For fractional changes of $\mathcal{T}_A$, the response in basal melt is more similar: a 10 % increase of $\mathcal{T}_A$ yields an increase in $M$ of about 15 % at both glaciers. (Note that a 10 % increase of $\mathcal{T}_A$ corresponds to an increase of $\sim 0.3$ and $\sim 0.4$ °C of $T_A$ at Ryder and 79°N, respectively.) In the absence of hydraulic control, where $M \propto \mathcal{T}_A^2$, the corresponding increase in $M$ would be 20 %. Figure 12

shows the model-predicted dependence of Ryder basal melt on $T_A$ and $h$. Currently, the AW interface is about 30 m below the transition depth $h_L$ at which hydraulic control ceases, i.e. a lifting of AW interface by about 30 m would bring the flow into the melt controlled regime.

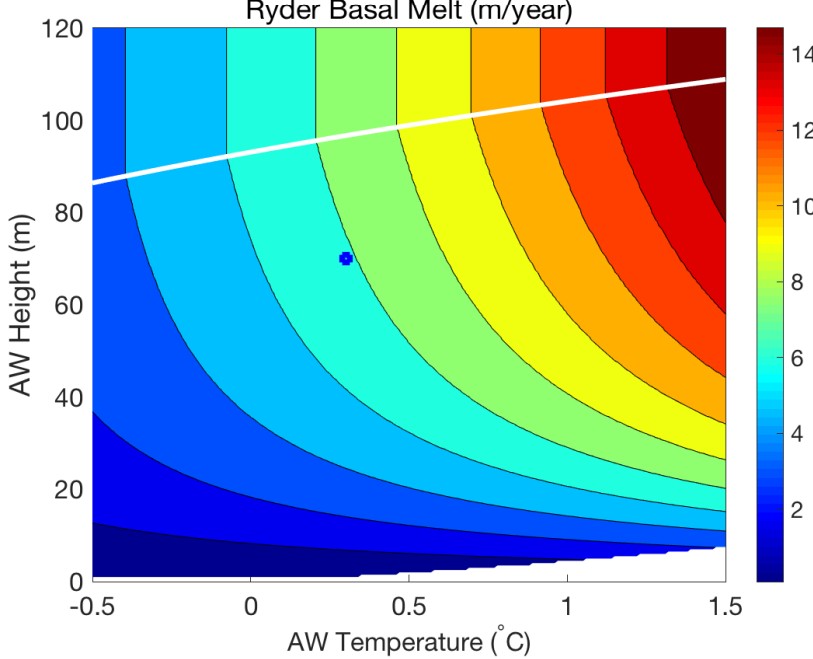

**Figure 12.** Estimated Ryder Glacier basal melt per unit area ($M/A$ given in meters per year) as a function of the AW temperature ($T_A$) and height ($h$) in model scenario 2; see section 3.2. Here, $n_1 = 2$ and $n_2 = 1$ and $M \propto \mathcal{T}^2$. The square indicates Ryder's present state, and the white line shows the transition between the melt- and hydraulically-controlled regimes. In the white section in the lower right-hand side of the figure, the temperature of water leaving the glacier ($T$) is below the freezing point, and refreezing is expected to occur: in this regime our melt representation needs to be modified.

At Petermann Glacier, the exchange flow is not hydraulically controlled because the height of the AW above the sill is larger than the transition height $h_L$ (Eq. 28). Note that $h_L$ decreases with the cross-sectional width of the sill ($h_L \propto W^{-2/3}$). Ryder and Petermann have similar values of $\mathcal{T}_A$, and it is primarily the wide sill in Petermann Fjord that yields a small value of $h_L$ (see Fig. 5 in Jakobsson et al., 2020).

## 4 Conclusions

To analyse the impact of hydraulic control on basal glacier melt, we have developed a two-layer fjord model that includes simple representations of melt and exchange-flow dynamics. Despite model idealisations, we believe that Figs. 6 and 7 qualitatively illustrate how the interplay between near-ice melt processes and hydraulic control affects the relationship between basal ice-tongue melt and AW features[2]. Our results suggest that there are two flow regimes, with different relationships between basal melt and AW features. To begin with, a melt-controlled flow regime, in which the fjord geometry and sills do not restrict the

---

[2]Figure 10 may be more representative for tidewater glaciers with subglacial discharge that exceeds subsurface ice melt.

exchange flow, and unmodified AW reaches the grounding line (Fig. 1). In this regime, the basal melt is rate limited by near-ice processes, rather than by the heat flux carried by the horizontal exchange flow in the fjord. Accordingly, the thermal forcing

is set by the AW temperature and the basal-melt processes determine the strength of the exchange flow. In this regime, the dependence of the basal melt on the AW height is weak and neglected here.

     In a sill fjord, hydraulic control can set an upper bound on the exchange flow, which depends on the height of the AW above the sill crest and the density difference between the in- and out-flowing waters. If hydraulic control is established, the outflow of glacially-modified water created by the basal melt must be compatible with the hydraulically-determined exchange flow at

the sill to ensure volume conservation. In a hydraulically-controlled flow regime, accordingly, the heat transport supplied by the fjord circulation enters as a rate-limiting factor for the basal melt (Fig. 1). The flow can transit from a melt-controlled to a hydraulically-controlled regime if the AW height is decreased or the AW temperature is increased (Figs. 6,7). Such transitions can also occur when increasing subglacial discharge enhances the basal melt and the production of glacially-modified water.

     In the hydraulic regime, decreasing AW height causes the exchange flow and its associated heat flux to decrease. As a

result, the basal melt decreases. The hydraulic constraint also reduces the sensitivity of the basal melt to changes of the AW temperature. We have examined this effect using a simple representation of the melt processes (Eqs. 15,16), and considered two idealised scenarios for how the flow adjusts to satisfy the hydraulic constraint; see sections 3.1 and 3.2. In these scenarios changes of the fjord stratification or entrainment of glacially-modified water into the inflowing AW are assumed to regulate the thermal forcing such that the hydraulic constraint is satisfied. In scenario 1 (Fig. 6), the thermal forcing is completely blind

to the AW temperature, but sensitive to the AW height. Scenario 2 (Fig. 7) is less extreme, and here the thermal forcing has a muted response to changes of the AW temperature, i.e. $\frac{d\mathcal{T}}{d\mathcal{T}_A} < 1$. These scenarios involve some fairly ad hoc assumptions, and further studies are needed to more accurately quantify the suppression of the thermal forcing in hydraulic flow regimes. Nevertheless, the qualitative features shown in Figs. 6 and 7 are expected to be robust. We note that scenario 2, which assumes entrainment and recirculation in the ice cavity, is more consistent with observations and modelling (Schaffer et al., 2020;

Jakobsson et al., 2020; Bao and Moffat, 2023) than scenario 1.

     The suppression of basal melt due to hydraulic control can be quantified by the reduction factor $R$ (Eq. 26): the melt relative to the case when AW reaches the grounding line is proportional to $R^{n_1}$. At an ice tongue or tidewater glacier in a specific fjord, $R$ is a function of the AW features, i.e. $R = R(\mathcal{T}_A, h)$. This feature could be used to parametrise effects of hydraulic control in simulations of marine-glacier response to changes of AW forcing, which is crucial for the evolution of Greenlandic marine

glaciers on decadal and centennial timescales (Straneo and Heimbach, 2013; Aschwanden et al., 2019; Wood et al., 2021).

     We have considered a situation where the sill is seaward of the ice-tongue front, which is presently the case for Ryder and 79°N glaciers. However, if an ice tongue extends above the sill, the ice draft will contribute to the geometrical constraints that determine the hydraulic exchange flow: the ice reduces the water-column depth over the sill. We will not explore this problem here. However, we note that in the early 1900s Ryder Ice Tongue was some 40 km longer than today, and covered the inner sill;

its front reached roughly to the 82°12'N mark in Fig. 2c; see Jakobsson et al. (2020) and O'Regan et al. (2021) for additional information. This should have strongly restricted the water exchange over the inner sill, resulting in very low basal melt on the inner part of the ice tongue.

Ryder Glacier has been relatively stable in recent decades (Hill et al., 2018). In contrast, Petermann Glacier, located ∼200 km southwest of Ryder, has been retreating and lost 35 km of its ice tongue in 2010 and 2012 (Johannessen et al., 2013; Hill et al., 2018). Jakobsson et al. (2020) proposed that Ryder Glacier has been stable because of its more restrictive sill geometry, which partly protects the ice tongue from inflow of warmer subsurface AW (Fig. 2). The present study suggests that Ryder has a relatively high $R$ value ($R \approx 0.9$), implying that despite the double sill geometry in Sherard Osborne Fjord, the modified AW reaching the grounding line is currently weakly cooled as its flows through the fjord. However, the sensitivity of basal melt to thermal forcing depends on local conditions such as ice-tongue geometry, subglacial discharge and tidal currents. Notably, our simple model fit (Table 2) suggests that the thermal sensitivity of basal melt per unit area ($\gamma_1/A$) is 40 % higher for Petermann than for Ryder and 79°N glaciers, which have comparable sensitivities. Further, remote sensing analyses show that the basal melt per unit area ($M/A$) is about 50 % higher on Petermann than on Ryder (Wilson et al., 2017). Even if our model fit is quite uncertain, this indicates that Ryder and 79°N glaciers, which are shielded by hydraulically-controlled sill flows, also have basal melt processes characterised by lower thermal sensitivity coefficients ($\gamma_1/A$) than Petermann.

We emphasise that the oceanic conditions at Ryder have only been observed a single time in the summer of 2019, and may not give a representative view of the melt–flow dynamic. As documented in the observations of Schaffer et al. (2020) at 79°N, variations of the AW height on monthly to annual timescales cause significant variations in exchange flow and basal melt. Thus, in hydraulically-controlled fjords, long-term melt variations may be strongly controlled by the evolution of the AW height, a quantity that has received less attention than the AW temperature for the evolution of marine glaciers in Greenland (Straneo and Heimbach, 2013; Wood et al., 2021). Central Arctic Ocean observations document changes, on decadal timescales, of the AW height that are up to 100 m (Polyakov et al., 2004). If similar height changes would occur along the arctic coast of Greenland, significant changes in basal glacial melt would result: the present model (scenario 2) suggests that a lowering of the AW height of ∼40 m would halve the basal melt on Ryder Ice Tongue.

*Data availability.* Data presented in the paper (multibeam bathymetry and oceanographic stations) are available in the Bolin Centre for Climate Research database. Multibeam bathymetry: https://doi.org/10.17043/ryder-2019-bathymetry. LADCP (current measurements): https://doi.org/10.170/ 2019-ladcp. CTD stations: https://doi.org/10.17043/ryder-2019-ctd.

**Appendix A: Subglacial discharge and conservations relations**

In the conservation relations of section 2.1, subglacial discharge (say $D$) is neglected, whereas it is allowed to affect the melt rate; see Eq. (15). Neglecting $D$ in the conservation relations is genrally not a valid approximation for tidewater glaciers, and we show here how to generalise the results to cases where the subglacial discharge is not small compared to the freshwater input due to subsurface ice melt ($M$). Essentially, this is accomplished by replacing $M$ by $M + D$ in the derivations presented in section 2.1, and this is outlined below.

When including $D$, conservation of volume is given by

$$Q = Q_A + M + D, \tag{A1}$$

but salt conservation is still given by Eq. (2). This yields the modified Knudsen's relation

$$\Delta S Q = S_A (M + D). \tag{A2}$$

The advective heat flux (Eq. 5) becomes

$$H = c[\Delta T Q + (T_f - T_A)(M + D)]. \tag{A3}$$

The advective heat flux determines the melt rate (Eq. 6), which in combination with Eq. (A3) yields

$$\Delta T Q = T_G M + (T_A - T_f)(M + D), \tag{A4}$$

where the Gade temperature $T_G$ is defined in Eq. (8). This equation is the modified form of Eq. (7). For conditions in North Greenland $(T_A - T_f)/T_G \approx 0.05$, implying that the last term in Eq. (A4) can be neglected unless $D \gg M$. This approximation is made in section 2.1, where $D$ is taken to be zero.

By combining Eqs. (A2,A4) and eliminating $Q$, we obtain the counterpart of Eq. (10):

$$\frac{\Delta S}{S_A} = \frac{\Delta T}{T_G} \Gamma; \tag{A5}$$

where we have introduced

$$\Gamma \overset{\text{def}}{=} \left( \frac{1}{1 + D/M} + \frac{T_A - T_f}{T_G} \right)^{-1}. \tag{A6}$$

Since $(T_A - T_f)/T_G < 1$, it follows that $\Gamma \geq 1$. The density difference (Eq. 12) as function of $\Delta T$ becomes

$$\frac{\Delta \rho}{\rho_0} = \frac{\Delta T}{T_G} (\beta S_A \Gamma - \alpha T_G). \tag{A7}$$

Thus for a given $\Delta T$, the primary effect of subglacial discharge is to increase the associated $\Delta S$ and $\Delta \rho$.

Figure A1 shows that depending on the value of $D/M$, there are two limiting regimes:

1. When $D/M \ll 1$, $\Gamma \approx 1$. Formally, this is the limit considered in section 2.1, where $D/M$ is taken to be zero. However, Fig. A1 indicates that this limit may serve as a leading order approximation also when $D/M \approx 1$.

2. When $D/M \gg 1$, $\Gamma \approx T_G/(T_A - T_f)$ ($\approx 20$ for conditions in North Greenland). This implies that $\Delta S/S_A \approx \Delta T/(T_A - T_f)$, which is the relationship between salinity and temperature changes when freshwater at the freezing temperature is mixed with AW. Here, $\beta S_A \Gamma \gg \alpha T_G$, and from Eqs. (A5,A7) it follows that the density difference becomes approximately controlled by the salinity difference alone: $\Delta \rho/\rho_0 \approx \beta \Delta S$. This limit is approached when $D/M$ is large compared to $T_G/(T_A - T_f)$, and can be appropriate for tidewater glaciers with high subglacial discharge.

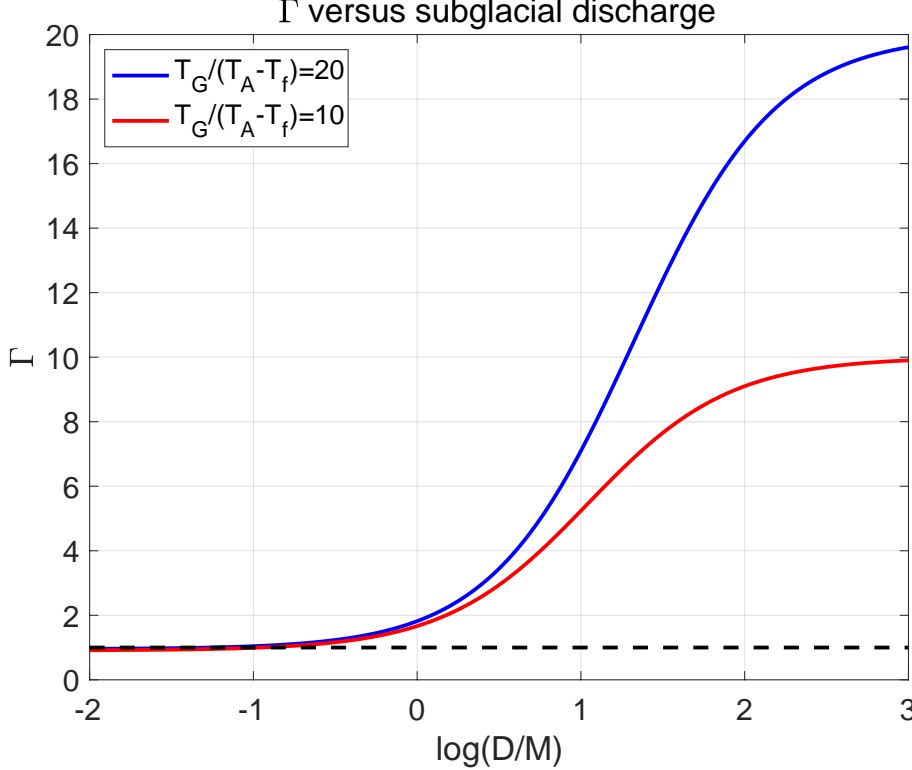

**Figure A1.** The factor $\Gamma$ (Eq. A6) as a function of the ratio between subglacial discharge and subsurface ice melt ($D/M$). The $x$-axis shows the logarithm ($\log_{10}$) of $D/M$, and the dashed black line shows $\Gamma = 1$. The blue line shows a case with $T_G/(T_A - T_f) = 20$, which is representative for conditions in northern Greenland, and the red line shows $T_G/(T_A - T_f) = 10$, characterising a case with warmer subsurface AW.

As long as the volume flux of subglacial discharge and basal melt is small compared to the fjord exchange flow (the generally valid case for Greenlandic fjords where $Q \gg M + D$) the inclusion of the factor $\Gamma$ in the generalised Eqs. (A5,A7) are the only model modification needed for treating cases where the subglacial discharge is not small compared to the melt. The flow in the melt-controlled regime (sec. 2.3) does not depend $\Delta S$ and $\Delta \rho$ and is therefore not dependent on the value of $\Gamma$. To describe the flow hydraulically-controlled regime (sec. 2.4), it is convenient to define a modified hydraulic coefficient

$$\tilde{k}_H \stackrel{\text{def}}{=} k_H \left( \frac{\beta S_A \Gamma - \alpha T_G}{\beta S_A - \alpha T_G} \right)^{1/2}, \tag{A8}$$

where $k_H$ is defined in Eq. (24) and we note that $\tilde{k}_H \geq k_H$. By replacing $k_H$ with $\tilde{k}_H$ in section 2.4 allow us the describe cases with high subglacial discharge.

For given AW features associated with a specific $\Delta T$, the primary effect of subglacial discharge (besides increasing the subsurface melt) is to enhance the layer density difference. Essentially, this causes the transition into the hydraulically-controlled

regime to occur for somewhat greater sills heights (or lower height of the AW layer) than when $D/M$ is taken to be zero. The regime transition is still defined from the condition that $Q_P = Q_H$ [see Eq. (27)], which yields the equivalent of Eq. (28):

$$h_L = \tilde{k}_H^{-2/3} \gamma_1^{-1/3} \gamma_2 \mathcal{T}_A^{\frac{3n_2 - n_1}{3}}, \tag{A9}$$

where $k_H$ is replaced $\tilde{k}_H$. This shows that $h_L$ (the height of the AW layer above the sill for which the flow becomes hydraulically controlled) decreases when $\Gamma$ increases $\tilde{k}_H$. In the limits when $D/M \ll 1$ or $D/M \gg 1$, $\Gamma$ and therefore $\tilde{k}_H$ are constants independent of $M$ and $D$. Accordingly, the results for the limit of small subglacial discharge ($D/M \ll 1$) of the present paper are qualitatively similar to those in the limit of high subglacial discharge ($D/M \gg 1$).

The situation is slightly more complicated between these two limits, where $D \sim M$. This is because $\Gamma$ and $\tilde{k}_H$ in this regime depend both on $M$ and $D$, which yields a model that is more complex algebraically. However, the main effect of finite values of $D/M$ is still to decrease the transition height $h_L$. Thus, the model results should qualitatively describe also cases where $D \sim M$.

## Appendix B: Mathematical relationships for scenario 2

Here, we derive a few mathematical relationships that can be used to construct graphs for scenario 2. We assume that $n_1 - n_2 = 1$, which simplifies the algebra but is not strictly necessary.

When $\mathcal{T}_A$ is fixed, it is convenient to put Eq. (43) in non-dimensional form using the variables

$$\tilde{\mathcal{T}} \stackrel{\text{def}}{=} \frac{\mathcal{T}}{\mathcal{T}_A}, \qquad \tilde{h} \stackrel{\text{def}}{=} \frac{h}{h_L}, \tag{B1}$$

where $h_L$ is defined in Eq. (28). Note that $\tilde{\mathcal{T}} = R$; see Eq. (26). By using these non-dimensional variables and Eqs. (37,35), we obtain

$$\tilde{\mathcal{T}} = 1 - \sigma \frac{\tilde{\mathcal{T}}^{\frac{2n_1}{3}}}{\tilde{h}} \left[ 1 - \tilde{h} \tilde{\mathcal{T}}^{-(n_2 - n_1/3)} \right]. \tag{B2}$$

Here, the term in the square brackets on the right-hand side is $\Phi$, and $\sigma$ is defined by Eq. (20). From Eq. (B2), we can obtain $\tilde{h}$ as a function of $\tilde{\mathcal{T}}$

$$\tilde{h} = \frac{\sigma \tilde{\mathcal{T}}^{\frac{2n_1}{3}}}{1 - \tilde{\mathcal{T}} + \sigma \tilde{\mathcal{T}}}. \tag{B3}$$

It is generally not possible to find the inverse function $\tilde{\mathcal{T}} = \tilde{\mathcal{T}}(\tilde{h})$ in a closed analytical form, but Eq. (B3) allow us examine it graphically by plotting $\mathcal{T}$ versus $h$. By using Eqs. (37,B3), we can after some manipulations find the dependence of $\Delta T$ and $\Phi$ on $\tilde{\mathcal{T}}$:

$$\frac{\Delta T}{\mathcal{T}_A} = 1 - \tilde{\mathcal{T}} + \sigma \tilde{\mathcal{T}}, \tag{B4}$$

$$\Phi = \frac{1 - \tilde{\mathcal{T}}}{1 - \tilde{\mathcal{T}} + \sigma \tilde{\mathcal{T}}}. \tag{B5}$$

To examine the melt dynamics when $h$ is fixed, it is useful to rewrite Eq. (43) as

$$\mathcal{T}_A = \mathcal{T} + \Delta T \Phi, \tag{B6}$$

where now the terms on the right-hand side are known functions of $\mathcal{T}$ and $h$ specified by Eqs. (37,35). We put Eq. (B6) in non-dimensional form using the definitions of $\mathcal{T}_L$, $\Delta T_L$, and $h_L$:

$$\frac{\mathcal{T}_A}{\mathcal{T}_L} = \frac{\mathcal{T}}{\mathcal{T}_L} + \sigma \left( \frac{\mathcal{T}}{\mathcal{T}_L} \right)^{\frac{2n_1}{3}} \left[ 1 - \left( \frac{\mathcal{T}_L}{\mathcal{T}} \right)^{\frac{3n_2 - n1}{3}} \right]. \tag{B7}$$

By using Eq. (B7), we obtain after some straightforward calculations

$$\left( \frac{d\mathcal{T}}{dT_A} \right)_{\mathcal{T} = \mathcal{T}_A} = \frac{1}{1 + \sigma(3n_2 - n_1)/3}, \tag{B8}$$

which applies at the regime transition in the hydraulically-controlled regime. In the melt-controlled regime, $\frac{d\mathcal{T}}{dT_A} = 1$. Thus, the suppression of $\mathcal{T}$ relative to the AW thermal forcing ($\mathcal{T}_A$) is governed by the exponents $n_1$ and $n_2$, and $\sigma$. An inspection of Eq. (35) shows that $(3n_2 - n_1)/3$ determines how fast the entrainment increases with $\mathcal{T}$.

*Author contributions.* JN lead the work on the paper. All authors contributed to the writing.

*Competing interests.* The authors have no competing interests.

*Acknowledgements.* This work was supported by the Swedish Research Council and the Swedish National Space Agency. We thank Jonas Nycander, Jonathan Wiskandt, and Inga Koszalka for valuable comments on our work.

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
