# Peer review of "Hydraulic suppression of basal glacier melt in sill fjords"

_EGUsphere, 2022_

## Author Comment (AC1)

**Response to Reviewer 1**

We thank the reviewer for providing constructive and positive critique of our manuscript. The points raised are relevant and interesting. We also thank the reviewer for pointing out numerous typos. Below, we propose a response to the reviewer's comments.

**Substantial comments**

I wonder about the applicability of the model to systems where subglacial discharge is significant (by which I mean that the subglacial discharge flux is comparable to or greater than the basal melt flux). I feel that many (most?) marine-terminating glaciers without ice tongues are likely to fall into this category, at least in summer. This comment has two parts:
**(i)** On neglecting subglacial discharge in the freshwater budget (L89) – is this appropriate? Based on Table 2 the observational estimates for the basal melt flux, M, are 60 m3/s at Ryder, 300 m3/s at Petermann and 600 m3/s at 79N. Cai et al. (2017) suggest based on RACMO2.3 surface runoff that subglacial discharge at Petermann can reach over 1000 m3/s in summer, which would significantly exceed the M term that is accounted for in the freshwater budget. On the other hand, Schaffer et al. (2020) suggest that only 11% of freshwater leaving the 79N cavity is subglacial discharge, which would support neglecting subglacial discharge. I feel a few more sentences justifying this assumption are needed. Also, for glaciers without ice tongues, it is much more likely that the subglacial discharge will significantly exceed the submarine/basal melt flux, so is the model applicable to fjords with tidewater glaciers, as you say in L53, when the subglacial discharge is neglected in the freshwater budget?
**(ii)** On the choice of the exponents n1 and n2 (L142 and discussion shortly after). For systems with high subglacial discharge, the buoyancy of the plume can be dominated by the subglacial discharge, so that the plume volume flux (and plume velocity) becomes independent of the thermal forcing and scales only with the subglacial discharge raised to the power 1/3. Some studies that investigate this regime are Jenkins 2011 and Straneo and Cenedese (2015) ? see in particular Eqs. 7 and 8 of the latter study. This subglacial discharge-dominated case would have n2=0 and n1 would be close to 1. I think it would be great to mention this possibility when discussing values for n1 and n2. And, if the model is to be widely applicable across Greenland fjords, do the results change much if n2=0? Or is this already a sub-case of what you have presented? I appreciate that this might require a lot extra to look into properly and that is not what I am proposing – maybe just a short consideration of how n2=0 might change things.
Overall, this substantial comment is not really a criticism of the paper and doesn?t require major changes to address, but would be worth considering as I think it has a bearing on how widely applicable the model would be.

**Response to to substantial comments**

The reviewer is right in pointing out that our assumption that the freshwater source from subglacial discharge ($D$) is small compared to the freshwater source from basal melt ($M$) does not generally apply to tidewater glaciers in summer when surface melting is strong. Some modifications of the model are required to describe cases where the subglacial discharge is much greater than the basal melt ($M \ll D$). We will revise the manuscript to make this clear.

**Substantial comment (i): How large is $M/D$ on the large Greenlandic ice tongues?**

The reviewer asks relevantly if our assumption that $M \gg D$ is valid for the large ice tongues of 79°N, Ryder, and Petermann. Moored hydrographic and velocity observations from the 79°N Glacier, reported by Schaffer et al. (2020), suggest that in the annual mean the subglacial discharge constitutes only about 10% of the freshwater exported from the glacier , i.e. $D/M \approx 0.1$. In addition, hydrographic observations taken in drill holes on Petermann Ice Tongue in August 2015 suggest that the freshwater fractions due to subglacial discharge in the glacially-modified near-ice water column is less than 30% (see Fig. 5 in Washam et al., 2019). These observations suggest that $D/M \ll 1$ can serve as a leading-order model approximation for 79°N, Ryder, and Petermann.

However, estimates of surface summer melt in the upstream catchments of these ice tongues based on regional atmospheric modelling show that this seasonal freshwater source (an upper-bound on the summer subglacial discharge) can be a few times larger than the annual-mean freshwater source due to basal melt on the ice tongues. On the other hand, the basal melt increases with the summer increase of subglacial discharge. Using data reported in the literature on annual-mean basal melt and subglacial discharge (JJA surface melt values dived by 4) (Wilson et al., 2017; Schaffer et al., 2020; Stranne et al., 2021; Slater and Straneo, 2022), we obtain annual-mean $D/M$ values of about 0.8, 0.3, and 0.7, respectively, for Ryder, 79°N, and Petermann. The results of Cai et al. (2017), who modeled basal melt on Petermann Ice Tongue, suggest that in summer (JJA) $D/M \sim 1$ and in the annual-mean $D/M \sim 0.7$.

*Suggested response*: In the beginning of section 2, we briefly state that observational and modelling results suggest that $D/M \ll 1$ can serve as a leading-order approximation when examining how hydraulic constraints affect the melt dynamics on large ice tongues such as 79°N, Ryder, and Petermann.

**Substantial comment (i): validity of the conceptual model**

It is relatively straightforward to include subglacial discharge $D$ in the conservation relations for mass, salt, and heat that are derived in section 2.1 in the paper. The novel feature is that Eq. (10) in the paper – the relation between the difference in salinity ($\Delta S$) and the temperature ($\Delta T$) between the two layers –

depends on the subglacial discharge:

$$\frac{\Delta S}{S_A} = \frac{\Delta T}{S_G}\Gamma,$$ (1)

where we have introduced

$$\Gamma \stackrel{\text{def}}{=} \left(\frac{M}{M+D} + \frac{T_A - T_f}{T_G}\right)^{-1}.$$ (2)

By noting that $(T_A - T_f)/T_G \sim 0.05$, it follows that $\Gamma \geq 1$; and we can identify three limiting cases depending on the value of $D/M$:

1. When $D/M \ll 1$, $\Gamma \approx 1$. This is the limit considered in the paper.

2. When $D/M \sim 1$, $\Gamma \approx 1 + D/M$.

3. When $D/M \gg 1$, $\Gamma \approx T_G/(T_A - T_f)$. This implies that $\Delta S/S_A \approx \Delta T/(T_A - T_f)$, which is the relationship between salinity and temperature changes when freshwater at the freezing temperature is mixed with Atlantic Water. This limit is approached when $D/M \sim 50$ or greater, and should be appropriate for tidewater glaciers with high subglacial discharge.

As long the subglacial discharge is small compared to the exchange flow in a fjord ($Q$), the inclusion of the factor $\Gamma$ in Eq. (1) is the only modification of the model needed for treating cases where the subglacial discharge is not small compared to the melt. The primary effect of increasing $\Gamma$ is to strengthen the layer density difference. Essentially, this causes the transition into the hydraulically-controlled regime to occur for somewhat greater sills heights than in the limit where $D/M \ll 1$. These considerations indicate that the model presented in the paper is qualitatively correct also when $M \sim D$.

*Suggested response*: In the revision, we will include an appendix that derives Eq. (1) and discusses briefly how strong subglacial discharge qualitatively modify the model results. To replace Eq. (10) in the paper with Eq. (1) above, however, gives an algebraically more complex model. Accordingly, we will stick to the conservations relations given in 2.1, which are formally valid when $D/M \ll 1$.

**Substantial comment (ii): the choice of the exponents $n_1$ and $n_2$**

The reviewer suggests that the case of melt parametrisation with $n_1 = 1$ and $n_2 = 0$ is relevant for systems with high subglacial discharge, and that this case would be interesting to discuss. In fact, we consider this case (see Fig. 10), without mentioning that it may be relevant for systems with high subglacial discharge.

*Suggested response*: In the revision, we will discuss the results in Fig. 10 (c,d) in relation to systems with high subglacial discharge. We will mention that the model results are formally valid only $D/M$, with reference to the proposed new appendix.

**Minor comments**

L14 – the use of "marine ice" – I worry that this terminology could be a bit confusing. I'd suggest rephrasing using "marine-terminating glaciers".

We will revise as suggested.

L26 – Slater et al., 2022 recently argued that for some regions, the impact of increasing subglacial discharge on submarine melt has been as important as AW temperature – could be worth acknowledging here.

We propose the reformulate in the following way: *In Greenland, basal melt is sensitive to the AW temperature (Straneo and Heimbach, 2013), and increases in AW temperature and subglacial discharge have been the major drivers of the retreat of outlet glaciers in deep Greenlandic fjords since the mid 1990s (Wood et al., 2021; Slater and Straneo, 2022).*

L32 – "can stabilise marine glaciers" – I feel this statement is too certain for this point in the paper. Perhaps "has the potential to stabilise marine glaciers"?

We will revise as suggested.

L34 – either here or somewhere else appropriate, I think it would be worth acknowledging that processes other than hydraulic control can also modify AW between the shelf and the glacier – for example vertical mixing due to velocity shear even in the absence of a sill, or icebergs.

We propose the reformulate in the following way: *Numerous observations of sill flows demonstrate that the vertical mixing increases strongly when the flow becomes hydraulically controlled (Pratt and Whitehead, 2007), and Jakobsson et al. (2020) and Schaffer et al. (2020) show that as inflowing AW passes over the sills and descends on the landward slopes, it mixes with overlaying cold glacially-modified water.*

Fig. 2, panels b and c – it would be great to have a scale bar for these panels.

We will fix this.

Fig. 3 – it would be great to have the locations of these profiles shown on Fig. 2b and 2c

We will examine if this is compatible with readability of Fig. 3; if not we will – in the figure caption – refer to locations in Fig. 2.

L78 – it would be nice to finish off the introduction with a sentence that bridges into the next section. For example, "We now describe a two-layer model to investigate .."

We will do as suggested.

L122 (and a few other places) – it would be more consistent to refer to "Eq." instead of "relation"

We will follow this suggestion.

L129 – is the value of rho0 ever actually used in the model? Or does the density difference always get normalised by rho0 (e.g. Eq. 22), in which case there would be no need to assume a value for rho0.

This is correct and we will simply write "where $\rho_0$ is a constant reference density ..."

L198 – I don't quite follow why the exchange flow increases with deltaT when $n_1 - n_2 > 1$. From Eq. 21, don't we require $n_2/(n_1 - n_2) > 0$? Which would give $2n_2 - n_1 > 0$, but perhaps I am mistaken.

We have assumed that $n_1 \geq 0$ and $n_2 \geq 0$, but did not state that clearly. When $n_2 \geq 0$ it is simple to see that $n_1 - n_2 > 0$ is the relevant criteria. We will state that $n_1 \geq 0$ and $n_2 \geq 0$ when the exponents are introduced.

L200 – I think somewhere in this paragraph it would be appropriate to cite Zhao et al. (2021), which similarly looked at parameterising hydraulically-controlled transport (e.g. Eq. 17 in that paper).

This paper is very relevant. We will cite the paper around L200 and also on L45.

Fig. 4 – could you say how the axes are non-dimensionalised?

The axes are non-dimensionalised such that the AW height is one when the AW thermal forcing is one; we will state this. Equation (28) shows that non-dimensionalisation can be done by selecting an arbitrary scale for the AW thermal forcing $(\mathcal{T}_A)$, and define a non-dimensional $h$ that is one when non-dimensional $\mathcal{T}_A$ is one.

L315 – suggest adding "in the case n1=2 and n2=1" at the end of the first sentence

We will follow this suggestion.

Figs. 6 and 7 – it could be better not to use the jet colorscale

We think that this is partly an aesthetic matter, and the jet scale is preferred by a color blind coauthor.

Fig. 12 caption – "see the text for details" – did this mean to look at the text for details on the refreezing, or for details on the figure more generally? I took it to mean details on refreezing, and I think I didn?t see those, so perhaps revise.

We will revise the caption to make this clear.

L492 – suggest adding "unmodified" before "AW".

Good suggestion!

L506 – on the two idealised scenarios – can you speculate which might be more realistic? Scenario 2 feels more realistic to me because there is more a gradual transition from no entrainment to some entrainment, but perhaps we are not able to say yet.

Scenario 1, which assumes no entrainment, is extreme and less likely: the observations from Ryder and 79N show that entrainment occurs. Thus, scenario 2 is more realistic. This is also supported by the paper of Bao and Moffat (2023), which has recently been published in Cryosphere Discussion. They report an ocean-modelling study of glacial melt in a silled fjord, and their results support scenario 2. We will state this around L506 and also in in the beginning of section 3.

L536 – on Ryder and 79N having "basal melt processes that are less sensitive to thermal forcing than Petermann" – surely according to your model, the basal melt processes at all of the glaciers are equally sensitive to thermal forcing (because $M$ varies as $T^{m_1}$)? So is the higher melt rate at Petermann likely due to factors beyond thermal forcing (i.e. gamma1 in your equations), such as subglacial discharge or grounding line depth or basal slope?

Good point! We will change to: "*basal melt processes characterised by lower thermal sensitivity coefficients $\gamma_1/A$ than Petermann*".

L538 – suggest adding "at 79N" after Schaffer et al. (2020)

We will follow this suggestion.

**References**

Bao, W. and C. Moffat, 2023: Impact of shallow sills on heat transport and stratification regimes in proglacial fjords. *EGUsphere*.

Cai, C., E. Rignot, D. Menemenlis and Y. Nakayama, 2017: Observations and modeling of ocean-induced melt beneath Petermann Glacier Ice Shelf in northwestern Greenland. *Geophys. Res. Lett.*, **44**, 8396–8403.

Jakobsson, M., L. A. Mayer, J. Nilsson, C. Stranne, B. Calder, M. O'Regan, J. W. Farrell, T. M. Cronin, V. Brüchert, J. Chawarski, B. Eriksson, J. Fredriksson, L. Gemery, A. Glueder, F. A. Holmes, K. Jerram, N. Kirchner, A. Mix, J. Muchowski, A. Prakash, B. Reilly, B. Thornton, A. Ulfsbo, E. Weidner, H. Åkesson, T. Handl, E. Ståhl, L.-G. Boze, S. Reed, G. West and J. Padman, 2020: Ryder Glacier in northwest Greenland is shielded from warm Atlantic water by a bathymetric sill. *Communications Earth & Environment*, **1**(45).

Pratt, L. J. and J. A. Whitehead, 2007: *Rotating Hydraulics: Nonlinear Topographic Effects in the Ocean and Atmosphere*. Springer Verlag, first edition.

Schaffer, J., W. J. v. T. Kanzow, J. E. A. L. von Albedyll and D. H. Roberts, 2020: Bathymetry constrains ocean heat supply to Greenland's largest glacier tongue. *Nature Geoscience*, **13**, 227–231.

Slater, D. A. and F. Straneo, 2022: Submarine melting of glaciers in Greenland amplified by atmospheric warming. *Nature Geoscience*.

Straneo, F. and P. Heimbach, 2013: North Atlantic warming and the retreat of Greenland's outlet glaciers. *Nature*, **504**, 36–43.

Stranne, C., J. Nilsson, A. Ulfsbo, M. O'Regan, H. K. Coxall, L. Meire, J. Muchowski, L. A. Mayer, V. Brüchert, J. Fredriksson, B. Thornton, J. Chawarski, G. West, E. Weidner and M. Jakobsson, 2021: The climate sensitivity of northern Greenland fjords is amplified through sea-ice damming. *Communications Earth & Environment*, **2**(70).

Washam, P., K. W. Nicholls, A. Münchow and L. Padman, 2019: Summer surface melt thins Petermann Gletscher ice shelf by enhancing channelized basal melt. *Journal of Glaciology*, **65**(252), 662–674.

Wilson, N., F. Straneo and P. Heimbach, 2017: Satellite-derived submarine melt rates and mass balance (2011–2015) for Greenland's largest remaining ice tongues. *The Cryosphere*, **11**(6), 2773–2782.

Wood, M., E. Rignot, I. Fenty, L. An, A. Bjørk, M. van den Broeke, C. Cai, E. Kane, D. Menemenlis, R. Millan, M. Morlighem, J. Mouginot, B. Noël, B. Scheuchl, I. Velicogna, J. K. Willis and H. Zhang, 2021: Ocean forcing drives glacier retreat in Greenland. *Science Advances*, **7**(1).

---

## Author Comment (AC2)

**Response to Reviewer 2**

We thank the reviewer for providing constructive and positive critique of our manuscript. The points raised are relevant and interesting. We also thank the reviewer for pointing out numerous typos. Below, we propose a response to the reviewer's comments.

**General comments**

From my point of view, the manuscript is well written and the study is well prepared and with many of the limitations of the model considered. However, there are two general aspects that I would like to comment on.
**(i)** Regarding the content, I have missed several details about the physical settings of Peterman and Ryder glaciers (see specific comments below), as well as the entire description of 79N Glacier system. In order to understand the transition range to which each system is subject, it would also be convenient to report approximately how deep the SPW, the front of the ice tongue, the sill and the AW reach in each system. In the limitations of the model, I believe that the implications of not including subglacial discharge in the melting flow should be further developed. It would also be convenient to take into account that melt buoyant plumes can reach neutral buoyancy at different depths. If either of these depths is close to $h_L$, it could encourage a transition to the hydraulically-controlled regime. Even, if the plume NBD is reached deeper than the sill, the plume flux would be totally trapped in the ice cavity; there would be no exchange flux to the oceanward side of the sill (entrainment ratio of 1) and the waters from the ice cavity would turn colder and fresher without input of the AW.

*Suggested response:* We will give more information on the three glacial system at appropriate sections in the paper. We agree with the reviewer that the validity and consequences of neglecting subglacial discharge in the conservation relations (section 2.1) deserves a discussion. Reviewer 1 raises a similar point, and in response we propose to add a new appendix that discusses how subglacial discharge that is higher than basal melt affects the model results qualitatively; see below and also the response to Reviewer 1.

The reviewer's point that that melt buoyant plumes can reach neutral buoyancy at different depths is relevant, but we deem that our model needs to be further developed to examine this interesting issue. Thus, we leave this point for future studies.

(ii) Regarding the organization of the content, I would consider it appropriate to make some adjustments to facilitate the understanding of the study, although I also understand that each person must have their own style and I propose my comment as a suggestion. I think the most appropriate structure would be:

1. Introduction: The state of the art, motivation and objectives as they are (I would not include here the subsection about the Physical settings of the glacier-fjord systems).

We think that the structure suggested by the reviewer is good and logical. However, to follow this suggestion would entail a major revision of the paper. Therefore, we propose to keep the structure of the paper.

**Specific comments and typos**

L.18 - Basal melt relates to the melting occurring at the base of an ice body (either grounded or floating). However, basal melt here only refers to the floating part of the ice (ice shelves and ice tongues), but it doesn't to tidewater glaciers with vertical ice front, which are also marine-terminating glaciers.
Good point: we will use subsurface melt, rather than basal melt.
L.19 – The definition of grounding line given here is only valid for ice shelves and ice tongues. The grounding line feature is also present in tidewater glaciers with a nearly-vertical ice front, where no permanent floating ice exists.
We propose to add a footnote explaining what applies for tidewater glaciers.
L.20 – The effects of water column stratification on submarine melting was also reported on De Andrs et al. (2020).
This is relevant reference that we will cite here, and on L290.
L.46 'effects due Earth's rotation' – 'effects due to Earth's rotation'.
We will correct this.
Section 1.1 - As stated in L.52-53: 'the model results are discussed in relation to observations from... and 79N glacier'. However, no information of the physical settings of 79N glacier is provided within the whole manuscript. Is there any reason for this lack of information?
We decided to focus the comparative discussion on Petermann and Ryder because they are relatively close geographically, and hence have similar Atlantic Water conditions outside the fjords. However, we agree with the reviewer that it is relevant to provide some information on 79N as well.

We propose to start section 1.1 by giving some general facts concerning these three ice tongues in North Greenland, and point to papers that describe 79N (e.g. Wilson et al., 2017; Lindeman et al., 2020; Schaffer et al., 2020), before we begin the comparison between Ryder and Petermann.
- In order to contextualize the rate of frontal advancing/retreating and get a frame of reference for the sill influence on hydraulic control, could the author specify what the lengths and widths of these two glaciers and fjords are? and the depth range of the grounding lines and ice-tongue fronts? I can only found GL and front details for Ryder glacier in Fig.2 caption.

We will provide the information the reviewer asks for in the caption of Fig. 2, and maybe also in Table 2.

L.57 - It would be better to quantify 'a relatively deep and wide sill'.

We propose the write: 'a ∼400 m deep and ∼12 km wide sill '.

Fig.2 - In panel a), it would be nice to have the coordinates frame and the North arrow.

- In panels b) and c), a scale bar (and a more precise bathymetric colorbar) would help with the fjord and sill dimensions. It seems that the coordinate 630'W appearing on the left y-axis of pannel b) is a mistake. It would also be helpful to have in these pannels (b and c) the location of the CTD casts used in Fig. 3.

We will revise the Fig. 2 as suggested.

L.80 and 86 - Based on CTD observations, it would be helpful to give a thickness range of the two layers considered in the model, as well as the thickness of the surface-polar-waters layer (it could also be highlighted on Fig.3).

We will work on a revised version of Fig. 3 that indicates the model layer thicknesses; if needed we will make a separate figure showing this.

What are the limitations on the study (if any) of avoiding mixing between glacially modified and surface polar waters?

The important model constraint is that glacially modified does reach and mix with near surface waters that receive a strong input of surface runoff, as this freshwater source is not included in the model. Some mixing between glacially-modified plume water and the layer of surface polar water is expected to occur. This will cool and freshen the outflowing waters to some extent, but this process is neglected for simplicity.

L.82-83 - What are the limitations of neglecting subglacial discharge as a mechanism enhancing basal ice-tongue melting?

The main effect of subglacial discharge that is comparable to or larger than the basal melt is that Eq. (10) in the paper – the relation between the difference in salinity ($\Delta S$) and the temperature ($\Delta T$) between the two layers – becomes modified and depends on the subglacial discharge. Essentially, $\Delta S$ will for a given $\Delta T$ be larger than predicted by Eq. (10). This strengthen the layer density difference and causes the transition into the hydraulically-controlled regime to occur for somewhat greater sills heights than in the limit where subglacial discharge is neglected in the freshwater budget. We propose to include a new appendix that briefly describes some qualitative effects of finite subglacial discharge; see also the response to Reviewer 1 on this point.

L.82-83 Table 1 - Last row, in Gade temperature relation, change the 'equal symbol' by the 'almost-equal symbol' (as it appears in L.111 and L.116). Also, I am a bit confused, since it seems to be inconsistencies with the magnitude and units of this Gade temperature. A value of 80 K is given in Table 1 (which is consistent applying the proposed relation therein), but a value of 80 C is given in L.111. From other studies (e.g. Jenkins, 1999; Mankoff et al., 2016), this Gade temperature values are about -90 C. Could you, please, unravel this question?

The reviewer points to some unclarity in our definition of the Gade temperature $T_G$, which would be the drop in temperature of a unit volume of water from

which sensible heat is extracted to melt ice corresponding to a unit volume of liquid water. The equivalent ice temperature used by Jenkins (1999) is essentially $-T_G \cdot \frac{\rho_w}{\rho_i}$; where the factor $\rho_w/\rho_i$ emerges because we use unit volumes of liquid water in the definition of $T_G$. We will state this in the text. To avoid possible confusion related to the use of both the Kelvin and Celsius temperature scales, we will change to use only Celsius. Also, we will use the following slightly more accurate numerical values $L/c \approx 75$ °C and $T_G = L/c + c_i/c(T_f - T_i) \approx 80$ °C (where the ice temperature $T_i = -15$ °C).

L.111 - Modify L/c value/units according to my previous comment.
We will do this using °C.

L.198 - 'This show the' → 'This shows that the'
We will change this.

L.204-205 - Could also a significant subglacial discharge flux motivate this subcritical-to-critical transition? Answered in L.272-273.
OK.

L.206-207 - See also Hager et al. (2022), where a simple model is used to estimate the proportion of refluxed freshwater in a silled fjord and the potential impacts on submarine melting are discussed.
This is a relevant paper, and we will cite it in the beginning of section 3.2.1.

Foot note 1 - The word 'than' is repeated twice in the second line.
We will correct.

L.245 - Shouldn't it be $R < 1$ in the hydraulic regime, since $R = 1$ is reserved for the melt-controlled regime?
Correct! We will change this.

Fig.4 - What are the h and deltaT used to make axes non-dimensional?
We will explain the non-dimensionalisation in the caption.

L.289-290 - Increased stratification generated by strong surface melting has also been observed to dampen submarine melting in tidewater glacier-fjord systems (De Andrs et al., 2020).
We will cite this relevant paper.

L.292 - 'The reasoning above and suggest that' → 'The reasoning above suggests that'.
We will correct this.

Fig.8 - in L.3, 'is smaller (greater) than one the hydraulic' → 'is smaller (greater) than one in the hydraulic'.
We will correct this.

L.446 - I understand the near-bottom temperatures for the ice cavity, to get better estimates of those temperatures near the grounding line, but, shouldn't outside-forced temperatures be those at the near-sill depth, where the flow exchanges are taken place?.
Figure shows that the near bottom temperatures outside the sills, which characterise AW temperatures, are essentially equal to temperature at the sill depth. We will rewrite to make this clear.

L.460 - Please, quantify 'with large error bars'. We removed 'with large error bars' as there is difficulty to provide such from measurements taken at one particular time. Instead we will write: ' to be on the order of $50 \cdot 10^3$ m$^3$ s$^{-1}$.'

Fig.11 - in L.2, 'and 79' → 'and 79N'.

We will correct.

Fig. 12 - To get a more comprehensive understanding, it would be nice to have the squares for the three glaciers, not only for the Ryder glacier.

Regrettably, this is not possible as the basal melt shown in Fig. 12 is based on the Ryder model parameters. Separate figures are needed to show the other glaciers, and we decided to present only the Ryder case.

L.490 - 'Our results suggests' → 'Our results suggest'.

We will correct.

L.521 - 'longer that today' → 'longer than today'.

We will correct.

**References**

Lindeman, M. R., F. Straneo, N. J. Wilson, J. M. Toole, R. A. Krishfield, N. L. Beaird, T. Kanzow and J. Schaffer, 2020: Ocean circulation and variability beneath Nioghalvfjerdsbræ (79 north glacier) ice tongue. *Journal of Geophysical Research: Oceans*, **125**(8), e2020JC016091.

Schaffer, J., W. J. v. T. Kanzow, J. E. A. L. von Albedyll and D. H. Roberts, 2020: Bathymetry constrains ocean heat supply to Greenland's largest glacier tongue. *Nature Geoscience*, **13**, 227–231.

Wilson, N., F. Straneo and P. Heimbach, 2017: Satellite-derived submarine melt rates and mass balance (2011–2015) for Greenland's largest remaining ice tongues. *The Cryosphere*, **11**(6), 2773–2782.

---

## Author Response (AR1)

**Point-by-point reply**

We again take to opportunity to thank the reviewers for insightful and constructive comments.

We have revised the paper following closely the response to the reviewers' comment that we posted in the on-line discussion. We will first describe how we have addressed some of the more substantial comments; our response to the detailed comments are listed below.

**Subglacial discharge**

In our conceptual model, we assume that the freshwater input due to subglacial discharge is small compared to the freshwater input due to subsurface ice melt. Both reviewers point out that this assumption is not generally valid for tidewater glaciers, and ask for a discussion of this point.

In response, we have added a new appendix (including a new figure) that derives the conservation relations presented in section 2.1 accounting for subglacial discharge. The appendix shows that the model results presented in the paper hold qualitatively also when the freshwater input due to subglacial discharge is large compared to the freshwater input due to subsurface melt. The regime of tidewater glaciers with high subglacial discharge deserves a separate study, but the present results should give some qualitative guidance of how hydraulic control affects subsurface melt dynamics also in this regime.

In the beginning of section 2, we refer to the results of the new appendix and also point to observational evidence suggesting that the limit of neglecting subglacial discharge can be a reasonable leading-order approximation for large Greenlandic ice tongues such as 79°N, Petermann, and Ryder.

We have also added a few sentences in section 3.2.5, stating that the case $n_1 = 1$ and $n_2 = 0$ may be relevant for tidewater glaciers. This statement is also included in the caption of Fig. 10.

**Information on fjords and glaciers: Figure 2 and 3**

The reviewers asked for some additional information on fjords and glaciers, particularly about 79°N, and offered suggestions for improving Figs. 2 and 3.

In response, we start section 1.1 by giving some general facts concerning Petermann, Ryder, 79°N glaciers in North Greenland, and point to papers that describe 79°N (e.g. Wilson et al., 2017; Lindeman et al., 2020; Schaffer et al., 2020), before we begin the comparison between Ryder and Petermann. We have also marked the location of 79°N in Fig. 2a.

For readability of Fig. 2, we decided to not indicate the positions of the CTD profiles shown in Fig. 3. In the caption of Fig. 3, we refer to Jakobsson et al. (2020) and Stranne et al. (2021) – where the positions of the CTD stations are given – and also refer back to Fig. 2 giving in words approximate positions of the stations.

We have revised Fig. 3 to indicate the vertical extents of the model's layers and sill depths for Petermann and Ryder.

**Minor comments reviewer 1**

L14 – the use of "marine ice" – I worry that this terminology could be a bit confusing. I'd suggest rephrasing using "marine-terminating glaciers".
We have revised as suggested.

L26 – Slater et al., 2022 recently argued that for some regions, the impact of increasing subglacial discharge on submarine melt has been as important as AW temperature – could be worth acknowledging here.
We have reformulated: *In Greenland, basal melt is sensitive to the AW temperature (Straneo and Heimbach, 2013), and increases in AW temperature and subglacial discharge have been the major drivers of the retreat of outlet glaciers in deep Greenlandic fjords since the mid 1990s (Wood et al., 2021; Slater and Straneo, 2022).*

L32 – "can stabilise marine glaciers" – I feel this statement is too certain for this point in the paper. Perhaps "has the potential to stabilise marine glaciers"?
We have revised as suggested.

L34 – either here or somewhere else appropriate, I think it would be worth acknowledging that processes other than hydraulic control can also modify AW between the shelf and the glacier – for example vertical mixing due to velocity shear even in the absence of a sill, or icebergs.
We have reformulated in the following way: *Numerous observations of sill flows demonstrate that the vertical mixing increases strongly when the flow becomes hydraulically controlled (Pratt and Whitehead, 2007), and Jakobsson et al. (2020) and Schaffer et al. (2020) show that as inflowing AW passes over the sills and descends on the landward slopes, it mixes with overlaying cold glacially-modified water.*

Fig. 2, panels b and c – it would be great to have a scale bar for these panels.
We have included scale bars.

Fig. 3 – it would be great to have the locations of these profiles shown on Fig. 2b and 2c
For readability of Fig. 2, we decided to not indicate the positions of the CTD profiles shown in Fig. 3. In the caption of Fig. 3, we refer to Jakobsson et al. (2020) and Stranne et al. (2021) – where the positions of the CTD stations are given – and also refers back to Fig. 2 describing approximate positions of the stations.

L78 – it would be nice to finish off the introduction with a sentence that bridges into the next section. For example, "We now describe a two-layer model to investigate .."
We have done as suggested.

L122 (and a few other places) – it would be more consistent to refer to "Eq." instead of "relation"
We have followed this suggestion.

L129 – is the value of rho0 ever actually used in the model? Or does the density difference always get normalised by rho0 (e.g. Eq. 22), in which case there would be no need to assume a value for rho0.
This is correct, and we simply write "where $\rho_0$ is a constant reference density ..."

L198 – I don't quite follow why the exchange flow increases with deltaT when $n_1 - n_2 > 1$. From Eq. 21, don't we require $n_2/(n_1 - n_2) > 0$? Which would give $2n_2 - n_1 > 0$, but perhaps I am mistaken.

We have assumed that $n_1 \geq 0$ and $n_2 \geq 0$, but did not state that clearly. When $n_2 \geq 0$ it is simple to see that $n_1 - n_2 > 0$ is the relevant criteria. We now state that $n_1 \geq 0$ and $n_2 \geq 0$ when the exponents are introduced.

L200 – I think somewhere in this paragraph it would be appropriate to cite Zhao et al. (2021), which similarly looked at parameterising hydraulically-controlled transport (e.g. Eq. 17 in that paper).

This paper is very relevant. We cite the paper around L200 and also on L45.

Fig. 4 – could you say how the axes are non-dimensionalised?

The axes are non-dimensionalised such that the AW height is one when the AW thermal forcing is one; we will state this. Equation (28) shows that non-dimensionalisation can be done by selecting an arbitrary scale for the AW thermal forcing ($\mathcal{T}_A$), and then define a non-dimensional $h$ that is one when non-dimensional $\mathcal{T}_A$ is one. This information is given in the figure caption.

L315 – suggest adding "in the case n1=2 and n2=1" at the end of the first sentence

Done.

Figs. 6 and 7 – it could be better not to use the jet colorscale

We think that this is partly an aesthetic matter, and the jet scale is preferred by a color blind coauthor.

Fig. 12 caption – "see the text for details" – did this mean to look at the text for details on the refreezing, or for details on the figure more generally? I took it to mean details on refreezing, and I think I didn't see those, so perhaps revise.

We have revised the figure caption to make this clear.

L492 – suggest adding "unmodified" before "AW".

Done.

L506 – on the two idealised scenarios – can you speculate which might be more realistic? Scenario 2 feels more realistic to me because there is more a gradual transition from no entrainment to some entrainment, but perhaps we are not able to say yet.

Scenario 1, which assumes no entrainment, is extreme and less likely: the observations from Ryder and 79N show that entrainment occurs. Thus, scenario 2 is more realistic. This is also supported by the paper of Bao and Moffat (2023), which has recently been published in Cryosphere Discussion. They report an ocean-modelling study of glacial melt in a silled fjord, and their results support scenario 2. We state this around L506 and also in in the beginning of section 3.

L536 – on Ryder and 79N having "basal melt processes that are less sensitive to thermal forcing than Petermann" – surely according to your model, the basal melt processes at all of the glaciers are equally sensitive to thermal forcing (because $M$ varies as $T^{n_1}$)? So is the higher melt rate at Petermann likely due to factors beyond thermal forcing (i.e. gamma1 in your equations), such as subglacial discharge or grounding line depth or basal slope?

Good point! We have changed to: "*basal melt processes characterised by lower thermal sensitivity coefficients $\gamma_1/A$ than Petermann*".

L538 – suggest adding "at 79N" after Schaffer et al. (2020)

Done.
Typos etc
Corrected.

**Specific comments and typos, reviewer 2**

L.18 - Basal melt relates to the melting occurring at the base of an ice body (either grounded or floating). However, basal melt here only refers to the floating part of the ice (ice shelves and ice tongues), but it doesn't to tidewater glaciers with vertical ice front, which are also marine-terminating glaciers.
We have changed to subsurface melt.
L.19 – The definition of grounding line given here is only valid for ice shelves and ice tongues. The grounding line feature is also present in tidewater glaciers with a nearly-vertical ice front, where no permanent floating ice exists.
We now write: "the point where the ice begins to float (or for tidewater glaciers, the water depth at their essentially vertical fronts)".
L.20 – The effects of water column stratification on submarine melting was also reported on De Andrs et al. (2020).
The paper is cited here, and on L290.
L.46 'effects due Earth's rotation' – 'effects due to Earth's rotation'.
Done.
Section 1.1 - As stated in L.52-53: 'the model results are discussed in relation to observations from... and 79N glacier'. However, no information of the physical settings of 79N glacier is provided within the whole manuscript. Is there any reason for this lack of information?
We start section 1.1 by giving some general facts concerning these three ice tongues in North Greenland, and point to papers that describe 79N (e.g. Wilson et al., 2017; Lindeman et al., 2020; Schaffer et al., 2020), before we begin the comparison between Ryder and Petermann.
- In order to contextualize the rate of frontal advancing/retreating and get a frame of reference for the sill influence on hydraulic control, could the author specify what the lengths and widths of these two glaciers and fjords are? and the depth range of the grounding lines and ice-tongue fronts? I can only found GL and front details for Ryder glacier in Fig.2 caption.
We now provide most of the information the reviewer asks for in the beginning of section 1.1, and in the captions of Fig. 2 and 3.
L.57 - It would be better to quantify 'a relatively deep and wide sill'.
We now write: 'a ∼400 m deep and ∼12 km wide sill '.
Fig.2 - In panel a), it would be nice to have the coordinates frame and the North arrow.
- In panels b) and c), a scale bar (and a more precise bathymetric colorbar) would help with the fjord and sill dimensions. It seems that the coordinate 630'W appearing on the left y-axis of pannel b) is a mistake. It would also be helpful to have in these pannels (b and c) the location of the CTD casts used in Fig. 3.

Figure 2 has been revised as suggested; but the CTD stations are not shown.

L.80 and 86 - Based on CTD observations, it would be helpful to give a thickness range of the two layers considered in the model, as well as the thickness of the surface-polar-waters layer (it could also be highlighted on Fig.3).

We have revised figure 3 to indicate the vertical extents of the model's layers.

What are the limitations on the study (if any) of avoiding mixing between glacially modified and surface polar waters?

The important model constraint is that glacially modified does reach and mix with near surface waters that receive a strong input of surface runoff, as this freshwater source is not included in the model. Some mixing between glacially-modified plume water and the layer of surface polar water is expected to occur. This will cool and freshen the outflowing waters to some extent, but this process is neglected for simplicity. We have not addressed this point.

L.82-83 - What are the limitations of neglecting subglacial discharge as a mechanism enhancing basal ice-tongue melting?

The main effect of subglacial discharge that is comparable to or larger than the basal melt is that Eq. (10) in the paper – the relation between the difference in salinity ($\Delta S$) and the temperature ($\Delta T$) between the two layers – becomes modified and depends on the subglacial discharge. Essentially, $\Delta S$ will for a given $\Delta T$ be larger than predicted by Eq. (10). This strengthen the layer density difference and causes the transition into the hydraulically-controlled regime to occur for somewhat greater sills heights than in the limit where subglacial discharge is neglected in the freshwater budget. We have included a new appendix that describes some qualitative effects of finite subglacial discharge; see response above.

L.82-83 Table 1 - Last row, in Gade temperature relation, change the 'equal symbol' by the 'almost-equal symbol' (as it appears in L.111 and L.116). Also, I am a bit confused, since it seems to be inconsistencies with the magnitude and units of this Gade temperature. A value of 80 K is given in Table 1 (which is consistent applying the proposed relation therein), but a value of 80 C is given in L.111. From other studies (e.g. Jenkins, 1999; Mankoff et al., 2016), this Gade temperature values are about -90 C. Could you, please, unravel this question?

The equivalent ice temperature used by Jenkins (1999) is essentially $-T_G \cdot \frac{\rho_w}{\rho_i}$; where the factor $\rho_w/\rho_i$ emerges because we use unit volumes of liquid water in the definition of $T_G$. We state this in the text. To avoid possible confusion related to the use of both the Kelvin and Celsius temperature scales, we use only Celsius. Also, we use the following slightly more accurate numerical values $L/c \approx 75$ °C and $T_G = L/c + c_i/c(T_f - T_i) \approx 80$ °C (where the ice temperature $T_i = -15$ °C).

L.111 - Modify L/c value/units according to my previous comment.

We have done this using °C.

L.198 - 'This show the' $\rightarrow$ 'This shows that the'

Done.

L.204-205 - Could also a significant subglacial discharge flux motivate this subcritical-to-critical transition? Answered in L.272-273.

OK.

L.206-207 - See also Hager et al. (2022), where a simple model is used to estimate the proportion of refluxed freshwater in a silled fjord and the potential impacts on submarine melting are discussed.

We cite this paper in the beginning of section 3.2.1.

Foot note 1 - The word 'than' is repeated twice in the second line.

Done.

L.245 - Shouldn't it be $R < 1$ in the hydraulic regime, since $R = 1$ is reserved for the melt-controlled regime?

Done.

Fig.4 - What are the h and deltaT used to make axes non-dimensional?

We explain the non-dimensionalisation in the caption.

L.289-290 - Increased stratification generated by strong surface melting has also been observed to dampen submarine melting in tidewater glacier-fjord systems (De Andrs et al., 2020).

We cite this relevant paper.

L.292 - 'The reasoning above and suggest that' $\rightarrow$ 'The reasoning above suggests that'.

We have corrected this.

Fig.8 - in L.3, 'is smaller (greater) than one the hydraulic' $\rightarrow$ 'is smaller (greater) than one in the hydraulic'.

Done.

L.446 - I understand the near-bottom temperatures for the ice cavity, to get better estimates of those temperatures near the grounding line, but, shouldn't outside-forced temperatures be those at the near-sill depth, where the flow exchanges are taken place?.

Figure 3 shows that the near bottom temperatures outside the sills, which characterise AW temperatures, are essentially equal to temperature at the sill depth. We have rewritten to make this clear.

L.460 - Please, quantify 'with large error bars'. We removed 'with large error bars' as there is difficulty to provide such from measurements taken at one particular time. Instead we write: ' to be on the order of $50 \cdot 10^3$ m$^3$ s$^{-1}$.'

Fig.11 - in L.2, 'and 79' $\rightarrow$ 'and 79N'.

Done.

Fig. 12 - To get a more comprehensive understanding, it would be nice to have the squares for the three glaciers, not only for the Ryder glacier.

Regrettably, this is not possible as the basal melt shown in Fig. 12 is based on the Ryder model parameters. Separate figures are needed to show the other glaciers, and we decided to present only the Ryder case.

L.490 - 'Our results suggests' $\rightarrow$ 'Our results suggest'.

Done.

L.521 - 'longer that today' $\rightarrow$ 'longer than today'.

Done.

**References**

Bao, W. and C. Moffat, 2023: Impact of shallow sills on heat transport and stratification regimes in proglacial fjords. *EGUsphere*.

Jakobsson, M., L. A. Mayer, J. Nilsson, C. Stranne, B. Calder, M. O'Regan, J. W. Farrell, T. M. Cronin, V. Brüchert, J. Chawarski, B. Eriksson, J. Fredriksson, L. Gemery, A. Glueder, F. A. Holmes, K. Jerram, N. Kirchner, A. Mix, J. Muchowski, A. Prakash, B. Reilly, B. Thornton, A. Ulfsbo, E. Weidner, H. Åkesson, T. Handl, E. Ståhl, L.-G. Boze, S. Reed, G. West and J. Padman, 2020: Ryder Glacier in northwest Greenland is shielded from warm Atlantic water by a bathymetric sill. *Communications Earth & Environment*, **1**(45).

Lindeman, M. R., F. Straneo, N. J. Wilson, J. M. Toole, R. A. Krishfield, N. L. Beaird, T. Kanzow and J. Schaffer, 2020: Ocean circulation and variability beneath Nioghalvfjerdsbræ (79 north glacier) ice tongue. *Journal of Geophysical Research: Oceans*, **125**(8), e2020JC016091.

Pratt, L. J. and J. A. Whitehead, 2007: *Rotating Hydraulics: Nonlinear Topographic Effects in the Ocean and Atmosphere*. Springer Verlag, first edition.

Schaffer, J., W. J. v. T. Kanzow, J. E. A. L. von Albedyll and D. H. Roberts, 2020: Bathymetry constrains ocean heat supply to Greenland's largest glacier tongue. *Nature Geoscience*, **13**, 227–231.

Slater, D. A. and F. Straneo, 2022: Submarine melting of glaciers in Greenland amplified by atmospheric warming. *Nature Geoscience*.

Straneo, F. and P. Heimbach, 2013: North Atlantic warming and the retreat of Greenland's outlet glaciers. *Nature*, **504**, 36–43.

Wilson, N., F. Straneo and P. Heimbach, 2017: Satellite-derived submarine melt rates and mass balance (2011–2015) for Greenland's largest remaining ice tongues. *The Cryosphere*, **11**(6), 2773–2782.

Wood, M., E. Rignot, I. Fenty, L. An, A. Bjørk, M. van den Broeke, C. Cai, E. Kane, D. Menemenlis, R. Millan, M. Morlighem, J. Mouginot, B. Noël, B. Scheuchl, I. Velicogna, J. K. Willis and H. Zhang, 2021: Ocean forcing drives glacier retreat in Greenland. *Science Advances*, **7**(1).

---

## Author Response (AR2)

Dear Jan De Rydt,

we are glad to hear that our manuscript has been accepted. We are thankful for the valuable comments of the reviewers, and we state this in the acknowledgment now.

We have addressed/corrected all the points you bring up, specifically:

Fig1 The colour of the $T_C$, $S_C$ water mass appears darker red than the $T_A$, $S_A$ water mass, and tends to (wrongly) suggest warmer conditions. Perhaps colours for $T_C$ and $T_A$ could be swapped?

We have changed colours in the sketch

L59 Is a subtitle needed here? Alternatively, the description of the geographic and oceanographic properties of these glaciers could become a short, separate section.

We made this part into a new section 2.

Figs 5-10: please consider adding (non-dim) and (units) to the axis and colorbar labels.

We have added (non dim) on axis, except for Fig 10; which would not read well (too crowded).

Best regards,

Johan Nilsson, and co-authors
Stockholm University